# Random Reshuffling: Simple Analysis with Vast Improvements

**Konstantin Mishchenko**
KAUST
Thuwal, Saudi Arabia

**Ahmed Khaled**
Cairo University
Giza, Egypt

**Peter Richtárik**
KAUST
Thuwal, Saudi Arabia

## Abstract

Random Reshuffling (RR) is an algorithm for minimizing finite-sum functions that utilizes iterative gradient descent steps in conjunction with data reshuffling. Often contrasted with its sibling Stochastic Gradient Descent (SGD), RR is usually faster in practice and enjoys significant popularity in convex and non-convex optimization. The convergence rate of RR has attracted substantial attention recently and, for strongly convex and smooth functions, it was shown to converge faster than SGD if 1) the stepsize is small, 2) the gradients are bounded, and 3) the number of epochs is large. We remove these 3 assumptions, improve the dependence on the condition number from $\kappa^2$ to $\kappa$ (resp. from $\kappa$ to $\sqrt{\kappa}$) and, in addition, show that RR has a different type of variance. We argue through theory and experiments that the new variance type gives an additional justification of the superior performance of RR. To go beyond strong convexity, we present several results for non-strongly convex and non-convex objectives. We show that in all cases, our theory improves upon existing literature. Finally, we prove fast convergence of the Shuffle-Once (SO) algorithm, which shuffles the data only once, at the beginning of the optimization process. Our theory for strongly convex objectives tightly matches the known lower bounds for both RR and SO and substantiates the common practical heuristic of shuffling once or only a few times. As a byproduct of our analysis, we also get new results for the Incremental Gradient algorithm (IG), which does not shuffle the data at all.

## 1 Introduction

We study the finite-sum minimization problem

$$\min_{x\in\mathbb{R}^d}\Big[f(x) = \tfrac{1}{n}\sum_{i=1}^{n} f_i(x)\Big], \tag{1}$$

where each $f_i : \mathbb{R}^d \to \mathbb{R}$ is differentiable and smooth, and are particularly interested in the big data machine learning setting where the number of functions $n$ is large. Thanks to their scalability and low memory requirements, first-order methods are especially popular in this setting (Bottou et al., 2018). *Stochastic* first-order algorithms in particular have attracted a lot of attention in the machine learning community and are often used in combination with various practical heuristics. Explaining these heuristics may lead to further development of stable and efficient training algorithms. In this work, we aim at better and sharper theoretical explanation of one intriguingly simple but notoriously elusive heuristic: *data permutation/shuffling*.

### 1.1 Data permutation

In particular, the goal of our paper is to obtain deeper theoretical understanding of methods for solving (1) which rely on random or deterministic *permutation/shuffling* of the data $\{1, 2, \ldots, n\}$ and

perform incremental gradient updates following the permuted order. We study three methods which belong to this class, described next.

An immensely popular but theoretically elusive method belonging to the class of data permutation methods is the **Random Reshuffling (RR)** algorithm (see Algorithm 1). This is the method we pay most attention to in this work, as reflected in the title. In each epoch $t$ of RR, we sample indices $\pi_0, \pi_1, \ldots, \pi_{n-1}$ *without replacement* from $\{1, 2, \ldots, n\}$, i.e., $\{\pi_0, \pi_1, \ldots, \pi_{n-1}\}$ is a random permutation of the set $\{1, 2, \ldots, n\}$, and proceed with $n$ iterates of the form

$$x_t^{i+1} = x_t^i - \gamma \nabla f_{\pi_i}(x_t^i),$$

where $\gamma > 0$ is a stepsize. We then set $x_{t+1} = x_t^n$, and repeat the process for a total of $T$ epochs. Notice that in RR, a *new* permutation/shuffling is generated at the beginning of each epoch, which is why the term *re*shuffling is used.

Furthermore, we consider the **Shuffle-Once (SO)** algorithm, which is identical to RR with the exception that it shuffles the dataset only once—at the very beginning—and then reuses this random permutation in all subsequent epochs (see Algorithm 2). Our results for SO follow as corollaries of the tools we developed in order to conduct a sharp analysis of RR.

Finally, we also consider the **Incremental Gradient (IG)** algorithm, which is identical to SO, with the exception that the initial permutation is not random but deterministic. Hence, IG performs incremental gradient steps through the data in a *cycling* fashion. The ordering could be *arbitrary*, e.g., it could be selected *implicitly* by the ordering the data comes in, or chosen *adversarially*. Again, our results for IG follow as a byproduct of our efforts to understand RR.

| **Algorithm 1** Random Reshuffling (RR) | **Algorithm 2** Shuffle Once (SO) |
|---|---|
| **Input:** Stepsize $\gamma > 0$, initial vector $x_0 = x_0^0 \in \mathbb{R}^d$, number of epochs $T$ | **Input:** Stepsize $\gamma > 0$, initial vector $x_0 = x_0^0 \in \mathbb{R}^d$, number of epochs $T$ |
| 1: **for** epochs $t = 0, 1, \ldots, T - 1$ **do** | 1: Sample a permutation $\pi_0, \pi_1, \ldots, \pi_{n-1}$ of $\{1, 2, \ldots, n\}$ |
| 2:     Sample a permutation $\pi_0, \pi_1, \ldots, \pi_{n-1}$ of $\{1, 2, \ldots, n\}$ | 2: **for** epochs $t = 0, 1, \ldots, T - 1$ **do** |
| 3:     **for** $i = 0, 1, \ldots, n - 1$ **do** | 3:     **for** $i = 0, 1, \ldots, n - 1$ **do** |
| 4:         $x_t^{i+1} = x_t^i - \gamma \nabla f_{\pi_i}(x_t^i)$ | 4:         $x_t^{i+1} = x_t^i - \gamma \nabla f_{\pi_i}(x_t^i)$ |
| 5:     $x_{t+1} = x_t^n$ | 5:     $x_{t+1} = x_t^n$ |

## 1.2   Brief literature review

RR is usually contrasted with its better-studied sibling Stochastic Gradient Descent (SGD), in which each $\pi_i$ is sampled uniformly *with replacement* from $\{1, 2, \ldots, n\}$. RR often converges faster than SGD on many practical problems (Bottou, 2009; Recht and Ré, 2013), is more friendly to cache locality (Bengio, 2012), and is in fact standard in deep learning (Sun, 2020).

The convergence properties of SGD are well-understood, with tightly matching lower and upper bounds in many settings (Rakhlin et al., 2012; Drori and Shamir, 2019; Nguyen et al., 2019). Sampling without replacement allows RR to leverage the finite-sum structure of (1) by ensuring that *each* function contributes to the solution once per epoch. On the other hand, it also introduces a significant complication: the steps are now *biased*. Indeed, in any iteration $i > 0$ within an epoch, we face the challenge of not having (conditionally) unbiased gradients since

$$\mathbb{E}\left[\nabla f_{\pi_i}(x_t^i) \mid x_t^i\right] \neq \nabla f(x_t^i).$$

This bias implies that individual iterations do not necessarily approximate a full gradient descent step. Hence, in order to obtain meaningful convergence rates for RR, it is necessary to resort to more involved proof techniques. In recent work, various convergence rates have been established for RR. However, a satisfactory, let alone complete, understanding of the algorithm's convergence remains elusive. For instance, the early line of attack pioneered by Recht and Ré (2012) seems to have hit the wall as their noncommutative arithmetic-geometric mean conjecture is not true (Lai and Lim, 2020).

The situation is even more pronounced with the SO method, as Safran and Shamir (2020) point out that there are no convergence results specific for the method, and the only convergence rates for SO follow by applying the worst-case bounds of IG. Rajput et al. (2020) state that a common practical heuristic is to use methods like SO that do not reshuffle the data every epoch. Indeed, they add that *"current theoretical bounds are insufficient to explain this phenomenon, and a new theoretical breakthrough may be required to tackle it"*.

IG has a long history owing to its success in training neural networks (Luo, 1991; Grippo, 1994), and its asymptotic convergence has been established early (Mangasarian and Solodov, 1994; Bertsekas and Tsitsiklis, 2000). Several rates for non-smooth & smooth cases were established by Nedić and Bertsekas (2001); Li et al. (2019); Gürbüzbalaban et al. (2019a); Ying et al. (2019) and Nguyen et al. (2020). Using IG poses the challenge of choosing a specific permutation for cycling through the iterates, which Nedić and Bertsekas (2001) note to be difficult. Bertsekas (2011) gives an example that highlights the susceptibility of IG to bad orderings compared to RR. Yet, thanks to Gürbüzbalaban et al. (2019b) and Haochen and Sra (2019), RR is known to improve upon both SGD and IG for *twice*-smooth objectives. Nagaraj et al. (2019) also study convergence of RR for smooth objectives, and Safran and Shamir (2020); Rajput et al. (2020) give lower bounds for RR and related methods.

## 2 Contributions

In this work, we study the convergence behavior of the data-permutation methods RR, SO and IG. While existing proof techniques succeed in obtaining insightful bounds for RR and IG, they fail to fully capitalize on the intrinsic power reshuffling and shuffling offers, and are not applicable to SO at all[1]. Our proof techniques are dramatically novel, simple, more insightful, and lead to improved convergence results, all under weaker assumptions on the objectives than prior work.

### 2.1 New and improved convergence rates for RR, SO and IG

In Section 3, we analyze the RR and SO methods and present novel convergence rates for strongly convex, convex, and non-convex smooth objectives. Our results for RR are summarized in Table 1.

- **Strongly convex case.** If each $f_i$ is strongly convex, we introduce a *new proof technique* for studying the convergence of RR/SO that allows us to obtain a *better dependence on problem constants*, such as the number of functions $n$ and the condition number $\kappa$, compared to prior work (see Table 1). Key to our results is a *new notion of variance specific to RR/SO* (see Definition 2), which we argue explains the superior convergence of RR/SO compared to SGD in many practical scenarios. Our result for SO tightly *matches the lower bound* of Safran and Shamir (2020). We prove similar results in the more general setting when each $f_i$ is convex and $f$ is strongly convex (see Theorem 2), but in this case we are forced to use smaller stepsizes.

- **Convex case.** For convex but not necessarily strongly convex objectives $f_i$, we give the first result showing that *RR/SO can provably achieve better convergence than SGD* for a large enough number of iterations. This holds even when comparing against results that assume second-order smoothness, like the result of Haochen and Sra (2019).

- **Non-convex case.** For non-convex objectives $f_i$, we obtain for RR a *much better dependence on the number of functions $n$* compared to the prior work of Nguyen et al. (2020).

Furthermore, in the appendix we formulate and prove convergence results for IG for strongly convex objectives, convex, and non-convex objectives as well. The bounds are worse than RR by a factor of $n$ in the noise/variance term, as IG does not benefit from randomization. Our result for strongly convex objectives *tightly matches the lower bound of Safran and Shamir (2020) up to an extra iteration and logarithmic factors, and is the first result to tightly match this lower bound*.

### 2.2 More general assumptions on the function class

Previous non-asymptotic convergence analyses of RR either obtain worse bounds that apply to IG, e.g., (Ying et al., 2019; Nguyen et al., 2020), or depend crucially on the assumption that each $f_i$ is Lipschitz (Nagaraj et al., 2019; Haochen and Sra, 2019; Ahn and Sra, 2020). Unfortunately, requiring each $f_i$

Table 1: Number of individual gradient evaluations needed by RR to reach an $\varepsilon$-accurate solution (defined in Section 3). Logarithmic factors and constants that are not related to the assumptions are ignored. For non-convex objectives, $A$ and $B$ are the constants given by Assumption 2.

| Assumptions | | $\mu$-Strongly Convex | Non-Strongly Convex | Non-Convex | Citation |
|---|---|---|---|---|---|
| N.L.[1] | U.V.[2] | | | | |
| ✓ | ✓ | $\kappa^2 n + \frac{\kappa n \sigma_*}{\mu\sqrt{\varepsilon}}$ | – | – | Ying et al. (2019) |
| ✗ | ✗ | $\kappa^2 n + \frac{\kappa\sqrt{n}G}{\mu\sqrt{\varepsilon}}$ | $\frac{LD^2}{\varepsilon} + \frac{G^2 D^2}{\varepsilon^2}$ [3] | – | Nagaraj et al. (2019) |
| ✗ | ✗ | – | – | $\frac{Ln}{\varepsilon^2} + \frac{LnG}{\varepsilon^3}$ | Nguyen et al. (2020) |
| ✓ | ✓ | $\frac{\kappa^2 n}{\sqrt{\mu\varepsilon}} + \frac{\kappa^2 n\sigma_*}{\mu\sqrt{\varepsilon}}$ [4] | – | – | Nguyen et al. (2020) |
| ✗ | ✗ | $\frac{\kappa\alpha}{\varepsilon^{1/\alpha}} + \frac{\kappa\sqrt{n}G\alpha^{3/2}}{\mu\sqrt{\varepsilon}}$ [5] | – | – | Ahn and Sra (2020) |
| ✓ | ✓ | $\kappa n + \frac{\sqrt{n}}{\sqrt{\mu\varepsilon}} + \frac{\kappa\sqrt{n}G_0}{\mu\sqrt{\varepsilon}}$ [6] | – | – | Ahn et al. (2020) |
| ✓ | ✓ | $\kappa + \frac{\sqrt{\kappa n}\sigma_*}{\mu\sqrt{\varepsilon}}$ [7]<br>$\kappa n + \frac{\sqrt{\kappa n}\sigma_*}{\mu\sqrt{\varepsilon}}$ | $\frac{Ln}{\varepsilon} + \frac{\sqrt{Ln}\sigma_*}{\varepsilon^{3/2}}$ | $\frac{Ln}{\varepsilon^2} + \frac{L\sqrt{n}(B+\sqrt{A})}{\varepsilon^3}$ | This work |

[1] Support for non-Lipschitz functions (N.L.): proofs without assuming that $\max_{i=1,\ldots,n} \|\nabla f_i(x)\| \leq G$ for all $x \in \mathbb{R}^d$ and some $G > 0$. Note that $\frac{1}{n}\sum_{i=1}^n \|\nabla f_i(x_*)\|^2 \overset{\text{def}}{=} \sigma_*^2 \leq G^2$ and $B^2 \leq G^2$.

[2] Unbounded variance (U.V.): there may be no constant $\sigma$ such that Assumption 2 holds with $A = 0$ and $B = \sigma$. Note that when the individual gradients are bounded, the variance is automatically bounded too.

[3] Nagaraj et al. (2019) require, for non-strongly convex functions, projecting at each iteration onto a bounded convex set of diameter $D$. We study the unconstrained problem.

[4] For strongly convex, Nguyen et al. (2020) bound $f(x) - f(x_*)$ rather than squared distances, hence we use strong convexity to translate their bound into a bound on $\|x - x_*\|^2$.

[5] The constant $\alpha > 2$ is a parameter to be specified in the stepsize used by (Ahn and Sra, 2020). Their full bound has several extra terms but we include only the most relevant ones.

[6] The result of Ahn et al. (2020) holds when $f$ satisfies the Polyak-Łojasiewicz inequality, a generalization of strong convexity. We nevertheless specialize it to strong convexity for our comparison. The constant $G_0$ is defined as $G_0 \overset{\text{def}}{=} \sup_{x:f(x)\leq f(x_0)} \max_{i\in[n]} \|\nabla f_i(x)\|$. Note that $\sigma_* \leq G_0$. We show a better complexity for PL functions under bounded variance in Theorem 4.

[7] This result is the first to show that RR and SO work with any $\gamma \leq \frac{1}{L}$, but it asks for each $f_i$ to be strongly convex. The second result assumes that only $f$ is strongly convex.

to be Lipschitz contradicts strong convexity (Nguyen et al., 2018) and is furthermore not satisfied in least square regression, matrix factorization, or for neural networks with smooth activations. In contrast, our work is the first to show how to leverage randomization to obtain better rates for RR without assuming each $f_i$ to be Lipschitz. In concurrent work, Ahn et al. (2020) also obtain a result for non-convex objectives satisfying the Polyak-Łojasiewicz inequality, a generalization of strong convexity. Their result holds without assuming bounded gradients or bounded variance, but unfortunately with a worse dependence on $\kappa$ and $n$ when specialized to $\mu$-strongly convex functions.

- **Strongly convex and convex case.** For strongly convex and convex objectives *we do not require any assumptions on the functions used beyond smoothness and convexity.*

- **Non-convex case.** For non-convex objectives we obtain our results under a significantly more general assumption than the bounded gradients assumptions employed in prior work. Our assumption is also provably satisfied when each function $f_i$ is lower bounded, and hence is *not only more general but also a more realistic assumption to use.*

# 3 Convergence theory

We will derive results for strongly convex, convex as well as non-convex objectives. To compare between the performance of first-order methods, we define an $\varepsilon$-accurate solution as a point $\tilde{x} \in \mathbb{R}^d$

that satisfies (in expectation if $\tilde{x}$ is random)

$$\|\nabla f(\tilde{x})\| \le \varepsilon, \qquad \text{or} \qquad \|\tilde{x} - x_*\|^2 \le \varepsilon, \qquad \text{or} \qquad f(\tilde{x}) - f(x_*) \le \varepsilon$$

for non-convex, strongly convex, and non-strongly convex objectives, respectively, and where $x_*$ is assumed to be a minimizer of $f$ if $f$ is convex. We then measure the performance of first-order methods by the number of individual gradients $\nabla f_i(\cdot)$ they access to reach an $\varepsilon$-accurate solution.

Our first assumption is that the objective is bounded from below and smooth. This assumption is used in all of our results and is widely used in the literature.

**Assumption 1.** The objective $f$ and the individual losses $f_1, \ldots, f_n$ are all $L$-smooth, i.e., their gradients are $L$-Lipschitz. Further, $f$ is lower bounded by some $f_* \in \mathbb{R}$. If $f$ is convex, we also assume the existence of a minimizer $x_* \in \mathbb{R}^d$.

Assumption 1 is necessary in order to obtain better convergence rates for RR compared to SGD, since without smoothness the SGD rate is optimal and cannot be improved (Nagaraj et al., 2019). The following quantity is key to our analysis and serves as an asymmetric distance between two points measured in terms of functions.

**Definition 1.** For any $i$, the quantity $D_{f_i}(x, y) \stackrel{\text{def}}{=} f_i(x) - f_i(y) - \langle \nabla f_i(y), x - y \rangle$ is the *Bregman divergence* between $x$ and $y$ associated with $f_i$.

It is well-known that if $f_i$ is $L$-smooth and $\mu$-strongly convex, then for all $x, y \in \mathbb{R}^d$

$$\tfrac{\mu}{2}\|x - y\|^2 \le D_{f_i}(x, y) \le \tfrac{L}{2}\|x - y\|^2, \tag{2}$$

so each Bregman divergence is closely related to the Euclidian distance. Moreover, the difference between the gradients of a convex and $L$-smooth $f_i$ is related to its Bregman divergence by

$$\|\nabla f_i(x) - \nabla f_i(y)\|^2 \le 2L \cdot D_{f_i}(x, y). \tag{3}$$

## 3.1 Main result: strongly convex objectives

Before we proceed to the formal statement of our main result, we need to present the central finding of our work. The analysis of many stochastic methods, including SGD, rely on the fact that the iterates converge to $x_*$ up to some noise. This is exactly where we part ways with the standard analysis techniques, since, it turns out, the intermediate iterates of shuffling algorithms converge to some other points. Given a permutation $\pi$, the real limit points are defined below,

$$x_*^i \stackrel{\text{def}}{=} x_* - \gamma \sum_{j=0}^{i-1} \nabla f_{\pi_j}(x_*), \qquad i = 1, \ldots, n-1. \tag{4}$$

In fact, it is predicted by our theory and later validated by our experiments that within an epoch the iterates *go away* from $x_*$, and closer to the end of the epoch they make a sudden comeback to $x_*$.

The second reason the vectors introduced in Equation (4) are so pivotal is that they allow us to define a new notion of variance. Without it, there seems to be no explanation for why RR sometimes overtakes SGD from the very beginning of optimization process. We define it below.

**Definition 2** (Shuffling variance). Given a stepsize $\gamma > 0$ and a random permutation $\pi$ of $\{1, 2, \ldots, n\}$, define $x_*^i$ as in (4). Then, the shuffling variance is given by

$$\sigma_{\text{Shuffle}}^2 \stackrel{\text{def}}{=} \max_{i=1,\ldots,n-1} \left[ \tfrac{1}{\gamma} \mathbb{E}\left[ D_{f_{\pi_i}}(x_*^i, x_*) \right] \right], \tag{5}$$

where the expectation is taken with respect to the randomness in the permutation $\pi$.

Naturally, $\sigma_{\text{Shuffle}}^2$ depends on the functions $f_1, \ldots, f_n$, but, unlike SGD, it also depends in a non-trivial manner on the stepsize $\gamma$. The easiest way to understand the new notation is to compare it to the standard definition of variance used in the analysis of SGD. We argue that $\sigma_{\text{Shuffle}}^2$ is the natural counter-part for the standard variance used in SGD. We relate both of them by the following upper and lower bounds:

**Proposition 1.** Suppose that each of $f_1, f_2, \ldots, f_n$ is $\mu$-strongly convex and $L$-smooth. Then $\frac{\gamma \mu n}{8}\sigma_*^2 \le \sigma_{\text{Shuffle}}^2 \le \frac{\gamma L n}{4}\sigma_*^2$, where $\sigma_*^2 \stackrel{\text{def}}{=} \frac{1}{n}\sum_{i=1}^{n} \|\nabla f_i(x_*)\|^2$.

In practice, $\sigma_{\mathrm{Shuffle}}^2$ may be much closer to the lower bound than the upper bound; see Section 4. This leads to a dramatic difference in performance and provides additional evidence of the superiority of RR over SGD. The next theorem states how exactly convergence of RR depends on the introduced variance.

**Theorem 1.** Suppose that the functions $f_1, \ldots, f_n$ are $\mu$-strongly convex and that Assumption 1 holds. Then for Algorithms 1 or 2 run with a constant stepsize $\gamma \leq \frac{1}{L}$, the iterates generated by either of the algorithms satisfy

$$\mathbb{E}\left[\|x_T - x_*\|^2\right] \leq (1 - \gamma\mu)^{nT}\|x_0 - x_*\|^2 + \frac{2\gamma\sigma_{\mathrm{Shuffle}}^2}{\mu}.$$

*Proof.* The key insight of our proof is that the intermediate iterates $x_t^1, x_t^2, \ldots$ do not converge to $x_*$, but rather converge to the sequence $x_*^1, x_*^2, \ldots$ defined by (4). Keeping this intuition in mind, it makes sense to study the following recursion:

$$
\begin{aligned}
\mathbb{E}\left[\|x_t^{i+1} - x_*^{i+1}\|^2\right] \\
= \mathbb{E}\left[\|x_t^i - x_*^i\|^2 - 2\gamma\langle\nabla f_{\pi_i}(x_t^i) - \nabla f_{\pi_i}(x_*), x_t^i - x_*^i\rangle + \gamma^2\|\nabla f_{\pi_i}(x_t^i) - \nabla f_{\pi_i}(x_*)\|^2\right]. \quad (6)
\end{aligned}
$$

Once we have this recursion, it is useful to notice that the scalar product can be decomposed as

$$
\begin{aligned}
\langle\nabla f_{\pi_i}(x_t^i) - \nabla f_{\pi_i}(x_*), x_t^i - x_*^i\rangle &= [f_{\pi_i}(x_*^i) - f_{\pi_i}(x_t^i) - \langle\nabla f_{\pi_i}(x_t^i), x_*^i - x_t^i\rangle] \\
&\quad + [f_{\pi_i}(x_t^i) - f_{\pi_i}(x_*) - \langle\nabla f_{\pi_i}(x_*), x_t^i - x_*\rangle] \\
&\quad - [f_{\pi_i}(x_*^i) - f_{\pi_i}(x_*) - \langle\nabla f_{\pi_i}(x_*), x_*^i - x_*\rangle] \\
&= D_{f_{\pi_i}}(x_*^i, x_t^i) + D_{f_{\pi_i}}(x_t^i, x_*) - D_{f_{\pi_i}}(x_*^i, x_*). \quad (7)
\end{aligned}
$$

This decomposition is, in fact, very standard and is a special case of the so-called *three-point identity* (Chen and Teboulle, 1993). So, it should not be surprising that we use it.

The rest of the proof relies on obtaining appropriate bounds for the terms in the recursion. Firstly, we bound each of the three Bregman divergence terms appearing in (7). By $\mu$-strong convexity of $f_i$, the first term in (7) satisfies

$$\frac{\mu}{2}\|x_t^i - x_*^i\|^2 \overset{(2)}{\leq} D_{f_{\pi_i}}(x_*^i, x_t^i),$$

which we will use to obtain contraction. The second term in (7) can be bounded via

$$\frac{1}{2L}\|\nabla f_{\pi_i}(x_t^i) - \nabla f_{\pi_i}(x_*)\|^2 \overset{(3)}{\leq} D_{f_{\pi_i}}(x_t^i, x_*),$$

which gets absorbed in the last term in the expansion of $\|x_t^{i+1} - x_*^{i+1}\|^2$. The expectation of the third divergence term in (7) is trivially bounded as follows:

$$\mathbb{E}\left[D_{f_{\pi_i}}(x_*^i, x_*)\right] \leq \max_{i=1,\ldots,n-1}\left[\mathbb{E}\left[D_{f_{\pi_i}}(x_*^i, x_*)\right]\right] = \gamma\sigma_{\mathrm{Shuffle}}^2.$$

Plugging these three bounds back into (7), and the resulting inequality into (6), we obtain

$$
\begin{aligned}
\mathbb{E}\left[\|x_t^{i+1} - x_*^{i+1}\|^2\right] &\leq \mathbb{E}\left[(1 - \gamma\mu)\|x_t^i - x_*^i\|^2 - 2\gamma(1 - \gamma L)D_{f_{\pi_i}}(x_t^i, x_*)\right] + 2\gamma^2\sigma_{\mathrm{Shuffle}}^2 \\
&\leq (1 - \gamma\mu)\mathbb{E}\left[\|x_t^i - x_*^i\|^2\right] + 2\gamma^2\sigma_{\mathrm{Shuffle}}^2. \quad (8)
\end{aligned}
$$

The rest of the proof is just solving this recursion, and is relegated to Section 8.2 in the appendix. ∎

We show (Corollary 1 in the appendix) that by carefully controlling the stepsize, the final iterate of RR after $T$ epochs satisfies

$$\mathbb{E}\left[\|x_T - x_*\|^2\right] = \tilde{\mathcal{O}}\left(\exp\left(-\frac{\mu nT}{L}\right)\|x_0 - x_*\|^2 + \frac{\kappa\sigma_*^2}{\mu^2 nT^2}\right), \quad (9)$$

where the $\tilde{\mathcal{O}}(\cdot)$ notation suppresses absolute constants and polylogarithmic factors. Note that Theorem 1 covers both RR and SO, and for SO, Safran and Shamir (2020) give almost the same *lower* bound. Stated in terms of the squared distance from the optimum, their lower bound is

$$\mathbb{E}\left[\|x_T - x_*\|^2\right] = \Omega\left(\min\left\{1, \frac{\sigma_*^2}{\mu^2 nT^2}\right\}\right),$$

where we note that in their problem $\kappa = 1$. This translates to sample complexity $\mathcal{O}\left(\sqrt{n}\sigma_*/(\mu\sqrt{\varepsilon})\right)$ for $\varepsilon \leq 1$[2]. Specializing $\kappa = 1$ in Equation (9) gives the sample complexity of $\tilde{\mathcal{O}}\left(1 + \sqrt{n}\sigma_*/(\mu\sqrt{\varepsilon})\right)$, matching the optimal rate up to an extra iteration. More recently, Rajput et al. (2020) also proved a similar lower bound for RR. We emphasize that Theorem 1 is not only tight, but it is also the first convergence bound that applies to SO. Moreover, it also immediately works if one permutes once every few epochs, which interpolates between RR and SO mentioned by Rajput et al. (2020).

**Comparison with SGD** To understand when RR is better than SGD, let us borrow a convergence bound for the latter. Several works have shown (e.g., see (Needell et al., 2014; Stich, 2019)) that for any $\gamma \leq \frac{1}{2L}$ the iterates of SGD satisfy

$$\mathbb{E}\left[\left\|x_{nT}^{\mathrm{SGD}} - x_*\right\|^2\right] \leq (1 - \gamma\mu)^{nT}\|x_0 - x_*\|^2 + \frac{2\gamma\sigma_*^2}{\mu}.$$

Thus, the question as to which method will be faster boils down to which variance is smaller: $\sigma_{\mathrm{Shuffle}}^2$ or $\sigma_*^2$. According to Proposition 1, it depends on both $n$ and the stepsize. Once the stepsize is sufficiently small, $\sigma_{\mathrm{Shuffle}}^2$ becomes smaller than $\sigma_*^2$, but this might not be true in general. Similarly, if we partition $n$ functions into $n/\tau$ groups, i.e., use minibatches of size $\tau$, then $\sigma_*^2$ decreases as $\mathcal{O}(1/\tau)$ and $\sigma_{\mathrm{Shuffle}}^2$ as $\mathcal{O}(1/\tau^2)$, so RR can become faster even without decreasing the stepsize. We illustrate this later with numerical experiments.

While Theorem 1 requires each $f_i$ to be strongly convex, we can also obtain results in the case where the individual strong convexity assumption is replaced by convexity. However, in such a case, we need to use a smaller stepsize, as the next theorem shows.

**Theorem 2.** Suppose that each $f_i$ is convex, $f$ is $\mu$-strongly convex, and Assumption 1 holds. Then provided the stepsize satisfies $\gamma \leq \frac{1}{\sqrt{2}Ln}$ the final iterate generated by Algorithms 1 or 2 satisfies

$$\mathbb{E}\left[\|x_T - x_*\|^2\right] \leq \left(1 - \frac{\gamma\mu n}{2}\right)^T\|x_0 - x_*\|^2 + \gamma^2\kappa n\sigma_*^2.$$

It is not difficult to show that by properly choosing the stepsize $\gamma$, the guarantee given by Theorem 2 translates to a sample complexity of $\tilde{\mathcal{O}}\left(\kappa n + \frac{\sqrt{\kappa n}\sigma_*}{\mu\sqrt{\varepsilon}}\right)$, which matches the dependence on the accuracy $\varepsilon$ in Theorem 1 but with $\kappa(n-1)$ additional iterations in the beginning. For $\kappa = 1$, this translates to a sample complexity of $\tilde{\mathcal{O}}\left(n + \frac{\sqrt{n}\sigma_*}{\mu\sqrt{\varepsilon}}\right)$ which is worse than the lower bound of Safran and Shamir (2020) when $\varepsilon$ is large. In concurrent work, Ahn et al. (2020) obtain in the same setting a complexity of $\tilde{\mathcal{O}}\left(1/\varepsilon^{1/\alpha} + \frac{\sqrt{n}G}{\mu\sqrt{\varepsilon}}\right)$ (for a constant $\alpha > 2$), which requires that each $f_i$ is Lipschitz and matches the lower bound only when the accuracy $\varepsilon$ is large enough that $1/\varepsilon^{1/\alpha} \leq 1$. Obtaining an optimal convergence guarantee for all accuracies $\varepsilon$ in the setting of Theorem 2 remains open.

### 3.2 Non-strongly convex objectives

We also make a step towards better bounds for RR/SO without any strong convexity at all and provide the following convergence statement.

**Theorem 3.** Let functions $f_1, f_2, \ldots, f_n$ be convex. Suppose that Assumption 1 holds. Then for Algorithm 1 or Algorithm 2 run with a stepsize $\gamma \leq \frac{1}{\sqrt{2}Ln}$, the average iterate $\hat{x}_T \overset{\text{def}}{=} \frac{1}{T}\sum_{j=1}^{T} x_j$ satisfies

$$\mathbb{E}\left[f(\hat{x}_T) - f(x_*)\right] \leq \frac{\|x_0 - x_*\|^2}{2\gamma nT} + \frac{\gamma^2 Ln\sigma_*^2}{4}.$$

Unfortunately, the theorem above relies on small stepsizes, but we still deem it as a valuable contribution, since it is based on a novel analysis. Indeed, the prior works showed that RR approximates a full gradient step, but we show that it is even closer to the implicit gradient step, see the appendix.

To translate the recursion in Theorem 3 to a complexity, one can choose a small stepsize and obtain (Corollary 2 in the appendix) the following bound for RR/SO:

$$\mathbb{E}\left[f(\hat{x}_T) - f(x_*)\right] = \mathcal{O}\left(\frac{L\|x_0 - x_*\|^2}{T} + \frac{L^{1/3}\|x_0 - x_*\|^{4/3}\sigma_*^{2/3}}{n^{1/3}T^{2/3}}\right).$$

Stich (2019) gives a convergence upper bound of $\mathcal{O}\left(\frac{L\|x_0-x_*\|^2}{nT} + \frac{\sigma_*\|x_0-x_*\|}{\sqrt{nT}}\right)$ for SGD. Comparing upper bounds, we see that RR/SO beats SGD when the number of epochs satisfies $T \geq \frac{L^2\|x_0-x_*\|^2 n}{\sigma_*^2}$. To the best of our knowledge, there are no strict lower bounds in this setting. Safran and Shamir (2020) suggest a lower bound of $\Omega\left(\frac{\sigma_*}{\sqrt{nT^3}} + \frac{\sigma_*}{nT}\right)$ by setting $\mu$ to be small in their lower bound for $\mu$-strongly convex functions, however this bound may be too optimistic.

## 3.3 Non-convex objectives

For non-convex objectives, we formulate the following assumption on the gradients variance.

**Assumption 2.** There exist nonnegative constants $A, B \geq 0$ such that for any $x \in \mathbb{R}^d$ we have,

$$\frac{1}{n}\sum_{i=1}^{n}\|\nabla f_i(x) - \nabla f(x)\|^2 \leq 2A\left(f(x) - f(x_*)\right) + B^2. \tag{10}$$

Assumption 2 is quite general: if there exists some $G > 0$ such that $\|\nabla f_i(x)\| \leq G$ for all $x \in \mathbb{R}^d$ and $i \in \{1, 2, \ldots, n\}$, then Assumption 2 is clearly satisfied by setting $A = 0$ and $B = G$. Assumption 2 also generalizes the uniformly bounded variance assumption commonly invoked in work on non-convex SGD, which is equivalent to (10) with $A = 0$. Assumption 2 is a special case of the Expected Smoothness assumption of Khaled and Richtárik (2020), and it holds whenever each $f_i$ is smooth and lower-bounded, as the next proposition shows.

**Proposition 2.** (Khaled and Richtárik, 2020, special case of Proposition 3) Suppose that $f_1, f_2, \ldots, f_n$ are lower bounded by $f_1^*, f_2^*, \ldots, f_n^*$ respectively and that Assumption 1 holds. Then there exist constants $A, B \geq 0$ such that Assumption 2 holds.

We now give our main convergence theorem for RR without assuming convexity.

**Theorem 4.** Suppose that Assumptions 1 and 2 hold. Then for Algorithm 1 run for $T$ epochs with a stepsize $\gamma \leq \min\left(\frac{1}{2Ln}, \frac{1}{(AL^2n^2T)^{1/3}}\right)$ we have

$$\min_{t=0,\ldots,T-1}\mathbb{E}\left[\|\nabla f(x_t)\|^2\right] \leq \frac{12(f(x_0)-f_*)}{\gamma nT} + 2\gamma^2 L^2 nB^2.$$

In the extended arxiv version, we also show that for PL functions satisfying Assumption 2 with $A = 0$ it holds $\mathbb{E}\left[f(x_t) - f_*\right] \leq \left(1 - \frac{\gamma\mu n}{2}\right)^T \left(f(x_0) - f_*\right) + \gamma^2 \kappa LnB^2$, where $\kappa = \frac{L}{\mu}$.

**Comparison with SGD.** From Theorem 4, one can recover the complexity that we provide in Table 1, see Corollary 3 in the appendix. Let's ignore some constants not related to our assumptions and specialize to uniformly bounded variance. Then, the sample complexity of RR, $K_{\mathrm{RR}} \geq \frac{L\sqrt{n}}{\varepsilon^2}(\sqrt{n} + \frac{\sigma}{\varepsilon})$, becomes better than that of SGD, $K_{\mathrm{SGD}} \geq \frac{L}{\varepsilon^2}(1 + \frac{\sigma^2}{\varepsilon^2})$, whenever $\sqrt{n}\varepsilon \leq \sigma$.

## 4 Experiments

We run our experiments on the $\ell_2$-regularized logistic regression problem given by

$$\frac{1}{N}\sum_{i=1}^{N}\left(-\left(b_i \log\left(h(a_i^\top x)\right) + (1-b_i)\log\left(1 - h(a_i^\top x)\right)\right)\right) + \frac{\lambda}{2}\|x\|^2,$$

where $(a_i, b_i) \in \mathbb{R}^d \times \{0, 1\}$, $i = 1, \ldots, N$ are the data samples and $h\colon t \to 1/(1 + e^{-t})$ is the sigmoid function. For better parallelism, we use minibatches of size 512 for all methods and datasets. We set $\lambda = L/\sqrt{N}$ and use stepsizes decreasing as $\mathcal{O}(1/t)$. See the appendix for more details on the parameters used, implementation details, and reproducibility.

**Reproducibility.** Our code is provided at https://github.com/konstmish/random_reshuffling. All used datasets are publicly available and all additional implementation details are provided in the appendix.

**Observations.** One notable property of all shuffling methods is that they converge with oscillations, as can be seen in Figure 1. There is nothing surprising about this as the proof of our Theorem 1 shows

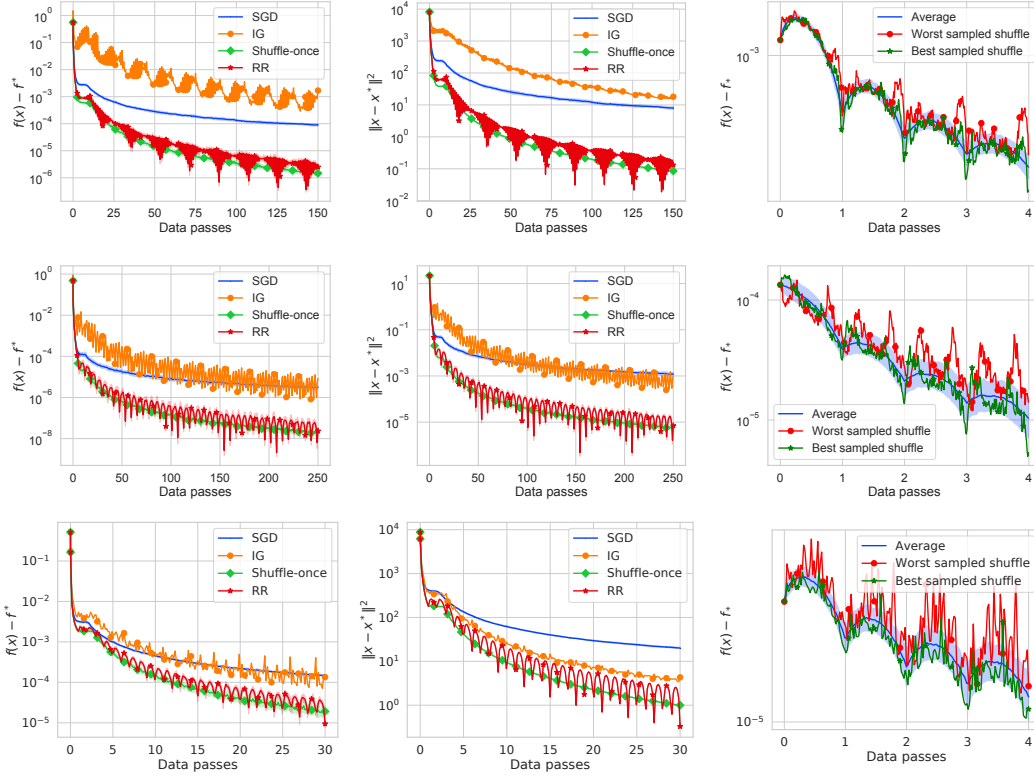

Figure 1: Top: `real-sim` dataset ($N = 72,309$; $d = 20,958$), middle row: `w8a` dataset ($N = 49,749$; $d = 300$), bottom: `RCV1` dataset ($N = 804,414$; $d = 47,236$). Left: convergence of $f(x_t^i)$, middle column: convergence of $\|x_t^i - x_*\|^2$, right: convergence of SO with different permutations .

that the intermediate iterates converge to $x_*^i$ instead of $x_*$. It is, however, surprising how striking the difference between the intermediate iterates within one epoch can be.

Next, one can see that SO and RR converge almost the same way, which is in line with Theorem 1. On the other hand, the contrast with IG is dramatic, suggesting existence of bad permutations. The probability of getting such a permutation seems negligible; see the right plot in Figure 2.

Finally, we remark that the first two plots in Figure 2 demonstrate the importance of the new variance introduced in Definition 2. The upper and lower bounds from Proposition 1 are depicted in these two plots and one can observe that the lower bound is often closer to the actual value of $\sigma_{\text{Shuffle}}^2$ than the upper bound. And the fact that $\sigma_{\text{Shuffle}}^2$ very quickly becomes smaller than $\sigma_*^2$ explains why RR often outperforms SGD starting from early iterations.

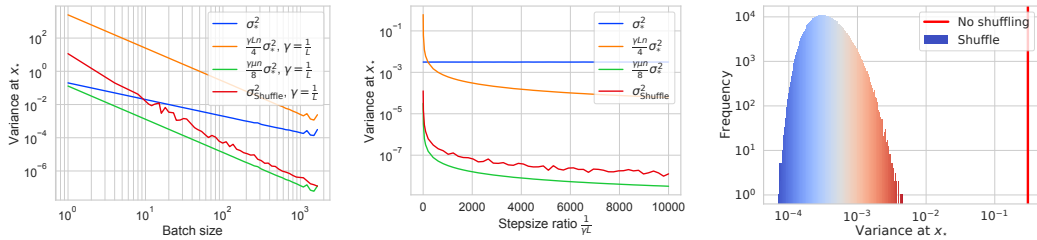

Figure 2: Estimated variance at the optimum, $\sigma_{\text{Shuffle}}^2$ and $\sigma_*^2$, for the `w8a` dataset. Left: the values of variance for different minibatch sizes with $\gamma = 1/L$. Middle: variance with fixed minibatch size 64 for different $\gamma$, starting with $\gamma = 1/L$ and ending with $\gamma = 10^{-4}/L$. Right: the empirical distribution of $\sigma_{\text{Shuffle}}^2$ for $500,000$ sampled permutations with $\gamma = 1/L$ and minibatch size 64.

## Broader Impact

Our contribution is primarily theoretical. Moreover, we study methods that are already in use in practice, but are notoriously hard to analyze. We believe we have made a breakthrough in this area by developing new and remarkably simple proof techniques, leading to sharp bounds. This, we hope, will inspire other researchers to apply and further develop our techniques to other contexts and algorithms. These applications may one day push the state of the art in practice for existing or new supervised machine learning applications, which may then have broader impacts. Besides this, we do not expect any direct or short term societal consequences.

## Acknowledgments and Disclosure of Funding

Ahmed Khaled acknowledges internship support from the Optimization and Machine Learning Lab led by Peter Richtárik at KAUST.

## Footnotes

[1]As we have mentioned before, the best known bounds for SO are those which apply to IG also, which means that the randomness inherent in SO is wholly ignored.

[2]In their problem, the initialization point $x_0$ satisfies $\|x_0 - x_*\|^2 \leq 1$ and hence asking for accuracy $\varepsilon > 1$ does not make sense.

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
