[Supplementary Material]

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

# Appendix

## Contents

## 5 Additional experiment details

**Objective properties.** To better correspond to the theoretical setting of our main result, we use $\ell_2$ regularization in every element of the finite-sum. To obtain minibatches for the RR, SO and IG we permute the dataset and then split it into $n = \lceil \frac{N}{\tau} \rceil$ groups of sizes $\tau, \ldots, \tau, N - \tau \left( \lceil \frac{N}{\tau} \rceil - 1 \right)$. In other words, the first $n-1$ groups are of size $\tau$ and the remaining samples go to the last group. For SO and IG, we split the data only once, and for RR, we do this at the beginning of each epoch. The permutation of samples used in IG is the one in which the datasets are stored online. The smoothness constant of the sum of logistic regression losses admits a closed form expression $L_f = \frac{1}{4N} \|A\|^2 + \lambda$. The individual losses are $L_{\max}$-smooth with $L_{\max} = \max_{i=1,\ldots,n} \|a_i\|^2 + \lambda$.

**Stepsizes.** For all methods in Figure 1, we keep the stepsize equal to $\frac{1}{L}$ for the first $k_0 = \lfloor K/40 \rfloor$ iterations, where $K$ is the total number of stochastic steps. This is important to ensure that there is an exponential convergence before the methods reach their convergence neighborhoods (Stich, 2019). After the initial $k_0$ iterations, the stepsizes used for RR, SO and IG were chosen as $\gamma_k = \min \left\{ \frac{1}{L}, \frac{3}{\mu \max\{1, k-k_0\}} \right\}$ and for SGD as $\gamma_k = \min \left\{ \frac{1}{L}, \frac{2}{\mu \max\{1, k-k_0\}} \right\}$. Although these stepsizes for RR are commonly used in practice (Bottou, 2009), we do not analyze them and leave decreasing-stepsize analysis for future work. We also note that although $L$ is generally not available, it can be estimated using empirically observed gradients (Malitsky and Mishchenko, 2019). For our experiments, we estimate $L$ of minibatches of size $\tau$ using the closed-form expressions from Proposition 3.8 in (Gower et al., 2019) as $L \leq \frac{n(\tau-1)}{\tau(n-1)} L_f + \frac{n-\tau}{\tau(n-1)} L_{\max}$. The confidence intervals in Figure 1 are estimated using 20 random seeds.

For the experiments in Figure 2, we estimate the expectation from (5) with 20 permutations, which provides sufficiently stable estimates. In addition, we use $L = L_{\max}$ (instead of using the batch smoothness of (Gower et al., 2019)) as the plots in this figure use different minibatch sizes and we want to isolate the effect of reducing the variance by minibatching from the effect of changing $L$.

**SGD implementation.** For SGD, we used two approaches to minibatching. In the first, we sampled $\tau$ indices from $\{1, \ldots, N\}$ and used them to form the minibatch, where $N$ is the total number of data samples. In the second approach, we permuted the data once and then at each iteration, we only sampled one index $i$ and formed the minibatch from indices $i, (i+1) \mod N, \ldots, (i+\tau-1) \mod N$. The latter approach is much more cash-friendly and runs significantly faster, while the iteration convergence was the same in our experiments. Thus, we used the latter option to produce the final plots.

For all plots and methods, we use zero initialization, $x_0 = (0, \ldots, 0)^\top \in \mathbb{R}^d$. We obtain the optimum, $x_*$, by running Nesterov's accelerated gradient method until it reaches machine precision. The plots in the right column in Figure 2 were obtained by initializing the methods at an intermediate iterate of Nesterov's method, and we found the average, best and worst results by sampling 1,000 permutations.

# 6 Basic facts and notation

## 6.1 Basic identities and inequalities

For any two vectors $a, b \in \mathbb{R}^d$, we have

$$2 \langle a, b \rangle = \|a\|^2 + \|b\|^2 - \|a - b\|^2. \tag{11}$$

As a consequence of (11), we get

$$\|a + b\|^2 \le 2\|a\|^2 + 2\|b\|^2. \tag{12}$$

**Convexity, strong convexity and smoothness.** A differentiable function $h : \mathbb{R}^d \to \mathbb{R}$ is called $\mu$-convex if for some $\mu \ge 0$ and for all $x, y \in \mathbb{R}^d$, we have

$$h(x) + \langle \nabla h(x), y - x \rangle + \frac{\mu}{2} \|y - x\|^2 \le h(y). \tag{13}$$

If $h$ satisfies (13) with $\mu > 0$, then we say that $h$ is $\mu$-strongly convex, and if $\mu = 0$ then we say $h$ is convex. A differentiable function $h : \mathbb{R}^d \to \mathbb{R}$ is called $L$-smooth if for some $L \ge 0$ and for all $x, y \in \mathbb{R}^d$, we have

$$\|\nabla h(x) - \nabla h(y)\| \le L \|x - y\|. \tag{14}$$

A useful consequence of $L$-smoothness is the inequality

$$h(x) \le h(y) + \langle \nabla h(y), x - y \rangle + \frac{L}{2} \|x - y\|^2, \tag{15}$$

holding for all $x, y \in \mathbb{R}^d$. If $h$ is $L$-smooth and lower bounded by $h_*$, then

$$\|\nabla h(x)\|^2 \le 2L \left( h(x) - h_* \right). \tag{16}$$

For any convex and $L$-smooth function $h$ it holds

$$D_h(x, y) \ge \frac{1}{2L} \|\nabla h(x) - \nabla h(y)\|^2. \tag{17}$$

**Jensen's inequality and consequences.** For a convex function $h : \mathbb{R}^d \to \mathbb{R}$ and any vectors $y_1, \ldots, y_n \in \mathbb{R}^d$, Jensen's inequality states that

$$h \left( \frac{1}{n} \sum_{i=1}^{n} y_i \right) \le \frac{1}{n} \sum_{i=1}^{n} h(y_i).$$

Applying this to the squared norm, $h(y) = \|y\|^2$, we get

$$\left\| \frac{1}{n} \sum_{i=1}^{n} y_i \right\|^2 \le \frac{1}{n} \sum_{i=1}^{n} \|y_i\|^2. \tag{18}$$

After multiplying both sides of (18) by $n^2$, we get

$$\left\| \sum_{i=1}^{n} y_i \right\|^2 \le n \sum_{i=1}^{n} \|y_i\|^2. \tag{19}$$

**Variance decomposition.** We will use the following decomposition that holds for any random variable $X$ with $\mathbb{E} \left[ \|X\|^2 \right] < +\infty$,

$$\mathbb{E} \left[ \|X\|^2 \right] = \|\mathbb{E} \left[ X \right]\|^2 + \mathbb{E} \left[ \|X - \mathbb{E} \left[ X \right]\|^2 \right]. \tag{20}$$

We will make use of the particularization of (20) to the discrete case: let $y_1, \ldots, y_m \in \mathbb{R}^d$ be given vectors and let $\bar{y} = \frac{1}{m} \sum_{i=1}^{m} y_i$ be their average. Then,

$$\frac{1}{m} \sum_{i=1}^{m} \|y_i\|^2 = \|\bar{y}\|^2 + \frac{1}{m} \sum_{i=1}^{m} \|y_i - \bar{y}\|^2. \tag{21}$$

Table 2: Summary of key notation used in the paper.

| Symbol | Description |
|---|---|
| $x_t$ | The iterate used at the start of epoch $t$. |
| $\pi$ | A permutation $\pi = (\pi_0, \pi_1, \ldots, \pi_{n-1})$ of $\{1, 2, \ldots, n\}$.<br>Fixed for Shuffle-Once and resampled every epoch for Random Reshuffling. |
| $\gamma$ | The stepsize used when taking descent steps in an epoch. |
| $x_t^i$ | The current iterate after $i$ steps in epoch $t$, for $0 \leq i \leq n$. |
| $g_t$ | The sum of gradients used over epoch $t$ such that $x_{t+1} = x_t - \gamma g_t$. |
| $\sigma_t^2$ | The variance of the individual loss gradients from the average loss at point $x_t$. |
| $A, B$ | Assumption 2 constants. |
| $L$ | The smoothness constant of $f$ and $f_1, f_2, \ldots, f_n$. |
| $\mu$ | The strong convexity constant (for strongly convex objectives). |
| $\kappa$ | The condition number $\kappa \overset{\text{def}}{=} L/\mu$ for strongly convex objectives. |
| $\delta_t$ | Functional suboptimality, $\delta_t = f(x_t) - f_*$, where $f_* = \inf_x f(x)$. |
| $r_t$ | The squared iterate distance from an optimum for convex losses<br>$r_t = \|x_t - x_*\|^2$. |
| $\mathbb{E}_t [\cdot]$ | Expectation conditional on the history of the algorithm prior to timestep $t$,<br>including $x_t$. |

## 6.2 Notation

We define the epoch total gradient $g_t$ as

$$g_t \overset{\text{def}}{=} \sum_{i=0}^{n-1} \nabla f_{\pi_i}(x_t^i).$$

We define the variance of the local gradients from their average at a point $x_t$ as

$$\sigma_t^2 \overset{\text{def}}{=} \frac{1}{n} \sum_{j=1}^{n} \|\nabla f_j(x_t) - \nabla f(x_t)\|^2.$$

By $\mathbb{E}_t [\cdot]$ we denote the expectation conditional on all information prior to iteration $t$, including $x_t$. To avoid issues with the special case $n = 1$, we use the convention $0/0 = 0$. A summary of the notation used in this work is given in Table 2.

## 7 A lemma for sampling without replacement

The following algorithm-independent lemma characterizes the variance of sampling a number of vectors from a finite set of vectors, without replacement. It is a key ingredient in our results on the convergence of the RR and SO methods.

**Lemma 1.** Let $X_1, \dots, X_n \in \mathbb{R}^d$ be fixed vectors, $\overline{X} \stackrel{\text{def}}{=} \frac{1}{n} \sum_{i=1}^{n} X_i$ be their average and $\sigma^2 \stackrel{\text{def}}{=} \frac{1}{n} \sum_{i=1}^{n} \left\| X_i - \overline{X} \right\|^2$ be the population variance. Fix any $k \in \{1, \dots, n\}$, let $X_{\pi_1}, \dots X_{\pi_k}$ be sampled uniformly without replacement from $\{X_1, \dots, X_n\}$ and $\overline{X}_\pi$ be their average. Then, the sample average and variance are given by

$$\mathbb{E}\left[\overline{X}_\pi\right] = \overline{X}, \qquad\qquad \mathbb{E}\left[\left\|\overline{X}_\pi - \overline{X}\right\|^2\right] = \frac{n-k}{k(n-1)}\sigma^2. \qquad (22)$$

*Proof.* The first claim follows by linearity of expectation and uniformity of sampling:

$$\mathbb{E}\left[\overline{X}_\pi\right] = \frac{1}{k}\sum_{i=1}^{k}\mathbb{E}\left[X_{\pi_i}\right] = \frac{1}{k}\sum_{i=1}^{k}\overline{X} = \overline{X}.$$

To prove the second claim, let us first establish that the identity $\mathrm{cov}(X_{\pi_i}, X_{\pi_j}) = -\frac{\sigma^2}{n-1}$ holds for any $i \neq j$. Indeed,

$$\mathrm{cov}(X_{\pi_i}, X_{\pi_j}) = \mathbb{E}\left[\left\langle X_{\pi_i} - \overline{X}, X_{\pi_j} - \overline{X}\right\rangle\right] = \frac{1}{n(n-1)}\sum_{l=1}^{n}\sum_{m=1, m\neq l}^{n}\left\langle X_l - \overline{X}, X_m - \overline{X}\right\rangle$$

$$= \frac{1}{n(n-1)}\sum_{l=1}^{n}\sum_{m=1}^{n}\left\langle X_l - \overline{X}, X_m - \overline{X}\right\rangle - \frac{1}{n(n-1)}\sum_{l=1}^{n}\left\| X_l - \overline{X}\right\|^2$$

$$= \frac{1}{n(n-1)}\sum_{l=1}^{n}\left\langle X_l - \overline{X}, \sum_{m=1}^{n}(X_m - \overline{X})\right\rangle - \frac{\sigma^2}{n-1}$$

$$= -\frac{\sigma^2}{n-1}.$$

This identity helps us to establish the formula for sample variance:

$$\mathbb{E}\left[\left\|\overline{X}_\pi - \overline{X}\right\|^2\right] = \frac{1}{k^2}\sum_{i=1}^{k}\sum_{j=1}^{k}\mathrm{cov}(X_{\pi_i}, X_{\pi_j})$$

$$= \frac{1}{k^2}\mathbb{E}\left[\sum_{i=1}^{k}\left\| X_{\pi_i} - \overline{X}\right\|^2\right] + \sum_{i=1}^{k}\sum_{j=1, j\neq i}^{k}\mathrm{cov}(X_{\pi_i}, X_{\pi_j})$$

$$= \frac{1}{k^2}\left(k\sigma^2 - k(k-1)\frac{\sigma^2}{n-1}\right) = \frac{n-k}{k(n-1)}\sigma^2. \qquad \blacksquare$$

# 8 Proofs for convex objectives (Sections 3.1 and 3.2)

## 8.1 Proof of Proposition 1

*Proof.* Let us start with the upper bound. Fixing any $i$ such that $1 \leq i \leq n-1$, we have $i(n-i) \leq \frac{n^2}{4} \leq \frac{n(n-1)}{2}$ and using smoothness and Lemma 1 leads to

$$
\mathbb{E}\left[D_{f_{\pi_i}}(x_*^i, x_*)\right] \overset{(15)}{\leq} \frac{L}{2}\mathbb{E}\left[\|x_*^i - x_*\|^2\right] = \frac{L}{2}\mathbb{E}\left[\left\|\sum_{k=0}^{i-1}\gamma\nabla f_{\pi_k}(x_*)\right\|^2\right]
$$

$$
\overset{(22)}{=} \frac{\gamma^2 L i(n-i)}{2(n-1)}\sigma_*^2
$$

$$
\leq \frac{\gamma^2 L n}{4}\sigma_*^2.
$$

To obtain the upper bound, it remains to take maximum with respect to $i$ on both sides and divide by $\gamma$. To prove the lower bound, we use strong convexity and the fact that $\max_i i(n-i) \geq \frac{n(n-1)}{4}$ holds for any integer $n$. Together, this leads to

$$
\max_i \mathbb{E}\left[D_{f_{\pi_i}}(x_*^i, x_*)\right] \overset{(13)}{\geq} \max_i \frac{\mu}{2}\mathbb{E}\left[\|x_*^i - x_*\|^2\right] = \max_i \frac{\gamma^2 \mu i(n-i)}{2(n-1)}\sigma_*^2 \geq \frac{\gamma^2 \mu n}{8}\sigma_*^2,
$$

as desired. ∎

## 8.2 Proof Remainder for Theorem 1

*Proof.* We start from (8) proved in the main text:

$$
\mathbb{E}\left[\|x_t^{i+1} - x_*^{i+1}\|^2\right] \leq (1-\gamma\mu)\mathbb{E}\left[\|x_t^i - x_*^i\|^2\right] + 2\gamma^2\sigma_{\text{Shuffle}}^2.
$$

Since $x_{t+1} - x_* = x_t^n - x_*^n$ and $x_t - x_* = x_t^0 - x_*^0$, we can unroll the recursion, obtaining the epoch level recursion

$$
\mathbb{E}\left[\|x_{t+1} - x_*\|^2\right] \leq (1-\gamma\mu)^n \mathbb{E}\left[\|x_t - x_*\|^2\right] + 2\gamma^2\sigma_{\text{Shuffle}}^2\left(\sum_{i=0}^{n-1}(1-\gamma\mu)^i\right).
$$

Unrolling this recursion across $T$ epochs, we obtain

$$
\mathbb{E}\left[\|x_T - x_*\|^2\right] \leq (1-\gamma\mu)^{nT}\|x_0 - x_*\|^2 + 2\gamma^2\sigma_{\text{Shuffle}}^2\left(\sum_{i=0}^{n-1}(1-\gamma\mu)^i\right)\left(\sum_{j=0}^{T-1}(1-\gamma\mu)^{nj}\right).
$$

(23)

The product of the two sums in (23) can be bounded by reparameterizing the summation as follows:

$$
\left(\sum_{j=0}^{T-1}(1-\gamma\mu)^{nj}\right)\left(\sum_{i=0}^{n-1}(1-\gamma\mu)^i\right) = \sum_{j=0}^{T-1}\sum_{i=0}^{n-1}(1-\gamma\mu)^{nj+i}
$$

$$
= \sum_{k=0}^{nT-1}(1-\gamma\mu)^k \leq \sum_{k=0}^{\infty}(1-\gamma\mu)^k = \frac{1}{\gamma\mu}.
$$

Plugging this bound back into (23), we finally obtain the bound

$$
\mathbb{E}\left[\|x_T - x_*\|^2\right] \leq (1-\gamma\mu)^{nT}\|x_0 - x_*\|^2 + 2\gamma\frac{\sigma_{\text{Shuffle}}^2}{\mu}.
$$

∎

## 8.3 Proof of complexity

In this subsection, we show how we get from Theorem 1 the complexity for strongly convex functions.

**Corollary 1.** Under the same conditions as those in Theorem 1, we choose stepsize

$$\gamma = \min\left\{\frac{1}{L}, \frac{2}{\mu nT}\log\left(\frac{\|x_0 - x_*\|\,\mu T\sqrt{n}}{\sqrt{\kappa}\sigma_*}\right)\right\}.$$

The final iterate $x_T$ then satisfies

$$\mathbb{E}\left[\|x_T - x_*\|^2\right] = \tilde{\mathcal{O}}\left(\exp\left(-\frac{\mu nT}{L}\right)\|x_0 - x_*\|^2 + \frac{\kappa\sigma_*^2}{\mu^2 nT^2}\right),$$

where $\tilde{O}(\cdot)$ denotes ignoring absolute constants and polylogarithmic factors. Thus, in order to obtain error (in squared distance to the optimum) less than $\varepsilon$, we require that the total number of iterations $nT$ satisfies

$$nT = \tilde{\Omega}\left(\kappa + \frac{\sqrt{\kappa n}\sigma_*}{\mu\sqrt{\varepsilon}}\right).$$

*Proof.* Applying Theorem 1, the final iterate generated by Algorithms 1 or 2 after $T$ epochs satisfies

$$\mathbb{E}\left[\|x_T - x_*\|^2\right] \le (1 - \gamma\mu)^{nT}\|x_0 - x_*\|^2 + 2\gamma\frac{\sigma_{\text{Shuffle}}^2}{\mu}.$$

Using Proposition 1 to bound $\sigma_{\text{Shuffle}}^2$, we get

$$\mathbb{E}\left[\|x_T - x_*\|^2\right] \le (1 - \gamma\mu)^{nT}\|x_0 - x_*\|^2 + \gamma^2\kappa n\sigma_*^2. \tag{24}$$

We now have two cases:

- **Case 1**: If $\frac{1}{L} \le \frac{2}{\mu nT}\log\left(\frac{\|x_0-x_*\|\mu T\sqrt{n}}{\sqrt{\kappa}\sigma_*}\right)$, then using $\gamma = \frac{1}{L}$ in (24) we have

$$\mathbb{E}\left[\|x_T - x_*\|^2\right] \le \left(1 - \frac{\mu}{L}\right)^{nT}\|x_0 - x_*\|^2 + \frac{\kappa n\sigma_*^2}{L^2}$$

$$\le \left(1 - \frac{\mu}{L}\right)^{nT}\|x_0 - x_*\|^2 + \frac{4\kappa\sigma_*^2}{\mu^2 nT^2}\log^2\left(\frac{\|x_0 - x_*\|\,\mu T\sqrt{n}}{\sqrt{\kappa}\sigma_*}\right).$$

Using that $1 - x \le \exp(-x)$ in the previous inequality, we get

$$\mathbb{E}\left[\|x_T - x_*\|^2\right] = \tilde{\mathcal{O}}\left(\exp\left(-\frac{\mu nT}{L}\right)\|x_0 - x_*\|^2 + \frac{\kappa\sigma_*^2}{\mu^2 nT^2}\right), \tag{25}$$

where $\tilde{\mathcal{O}}(\cdot)$ denotes ignoring polylogarithmic factors and absolute (non-problem specific) constants.

- **Case 2**: If $\frac{2}{\mu nT}\log\left(\frac{\|x_0-x_*\|\mu T\sqrt{n}}{\sqrt{\kappa}\sigma_*}\right) < \frac{1}{L}$, recall that by Theorem 1,

$$\mathbb{E}\left[\|x_T - x_*\|^2\right] \le (1 - \gamma\mu)^{nT}\|x_0 - x_*\|^2 + \gamma^2\kappa n\sigma_*^2. \tag{26}$$

Plugging in $\gamma = \frac{2}{\mu nT}\log\left(\frac{\|x_0-x_*\|\mu T\sqrt{n}}{\sqrt{\kappa}\sigma_*}\right)$, the first term in (26) satisfies

$$(1 - \gamma\mu)^{nT}\|x_0 - x_*\|^2 \le \exp\left(-\gamma\mu nT\right)\|x_0 - x_*\|^2$$

$$= \exp\left(-2\log\left(\frac{\|x_0 - x_*\|\,\mu T\sqrt{n}}{\sqrt{\kappa}\sigma_*}\right)\right)\|x_0 - x_*\|^2$$

$$= \frac{\kappa\sigma_*^2}{\mu^2 nT^2}. \tag{27}$$

Furthermore, the second term in (26) satisfies

$$\gamma^2\kappa n\sigma_*^2 = \frac{4\kappa\sigma_*^2}{\mu^2 nT^2}\log^2\left(\frac{\|x_0 - x_*\|\,\mu T\sqrt{n}}{\sqrt{\kappa}\sigma_*}\right). \tag{28}$$

Substituting (27) and (28) into (26), we get

$$\mathbb{E}\left[\|x_T - x_*\|^2\right] = \tilde{\mathcal{O}}\left(\frac{\kappa\sigma_*^2}{\mu^2 nT^2}\right). \tag{29}$$

This concludes the second case.

It remains to take the maximum of (25) from the first case and (29) from the second case. ∎

## 8.4 Two lemmas for Theorems 2 and 3

In order to prove Theorems 2 and 3, it will be useful to define the following quantity.

**Definition 3.** Let $x_t^0, x_t^1, \ldots, x_t^n$ be iterates generated by Algorithms 1 or 2. We define the forward per-epoch deviation over the $t$-th epoch $\mathcal{V}_t$ as

$$\mathcal{V}_t \overset{\text{def}}{=} \sum_{i=0}^{n-1} \left\| x_t^i - x_{t+1} \right\|^2. \tag{30}$$

We will now establish two lemmas. First, we will show that $\mathcal{V}_t$ can be efficiently upper bounded using Bregman divergences and the variance at the optimum. Subsequently use this bound to establish the convergence of RR/SO.

### 8.4.1 Bounding the forward per-epoch deviation

**Lemma 2.** Consider the iterates of Random Reshuffling (Algorithm 1) or Shuffle-Once (Algorithm 2). If the functions $f_1, \ldots, f_n$ are convex and Assumption 1 is satisfied, then

$$\mathbb{E}\left[\mathcal{V}_t\right] \leq 4\gamma^2 n^2 L \sum_{i=0}^{n-1} \mathbb{E}\left[D_{f_{\pi_i}}(x_*, x_t^i)\right] + \frac{1}{2}\gamma^2 n^2 \sigma_*^2, \tag{31}$$

where $\mathcal{V}_t$ is defined as in Definition 3, and $\sigma_*^2$ is the variance at the optimum given by $\sigma_*^2 \overset{\text{def}}{=} \frac{1}{n}\sum_{i=1}^{n}\left\|\nabla f_i(x_*)\right\|^2$.

*Proof.* For any fixed $k \in \{0, \ldots, n-1\}$, by definition of $x_t^k$ and $x_{t+1}$ we get the decomposition

$$x_t^k - x_{t+1} = \gamma \sum_{i=k}^{n-1} \nabla f_{\pi_i}(x_t^i) = \gamma \sum_{i=k}^{n-1}(\nabla f_{\pi_i}(x_t^i) - \nabla f_{\pi_i}(x_*)) + \gamma \sum_{i=k}^{n-1} \nabla f_{\pi_i}(x_*).$$

Applying Young's inequality to the sums above yields

$$
\begin{aligned}
\left\|x_t^k - x_{t+1}\right\|^2 &\overset{(12)}{\leq} 2\gamma^2 \left\|\sum_{i=k}^{n-1}(\nabla f_{\pi_i}(x_t^i) - \nabla f_{\pi_i}(x_*))\right\|^2 + 2\gamma^2 \left\|\sum_{i=k}^{n-1} \nabla f_{\pi_i}(x_*)\right\|^2 \\
&\overset{(19)}{\leq} 2\gamma^2 n \sum_{i=k}^{n-1}\left\|\nabla f_{\pi_i}(x_t^i) - \nabla f_{\pi_i}(x_*)\right\|^2 + 2\gamma^2 \left\|\sum_{i=k}^{n-1} \nabla f_{\pi_i}(x_*)\right\|^2 \\
&\overset{(17)}{\leq} 4\gamma^2 L n \sum_{i=k}^{n-1} D_{f_{\pi_i}}(x_*, x_t^i) + 2\gamma^2 \left\|\sum_{i=k}^{n-1} \nabla f_{\pi_i}(x_*)\right\|^2 \\
&\leq 4\gamma^2 L n \sum_{i=0}^{n-1} D_{f_{\pi_i}}(x_*, x_t^i) + 2\gamma^2 \left\|\sum_{i=k}^{n-1} \nabla f_{\pi_i}(x_*)\right\|^2.
\end{aligned}
$$

Summing up and taking expectations leads to

$$\sum_{k=0}^{n-1} \mathbb{E}\left[\left\|x_t^k - x_{t+1}\right\|^2\right] \leq 4\gamma^2 L n^2 \sum_{i=0}^{n-1} \mathbb{E}\left[D_{f_{\pi_i}}(x_*, x_t^i)\right] + 2\gamma^2 \sum_{k=0}^{n-1} \mathbb{E}\left[\left\|\sum_{i=k}^{n-1} \nabla f_{\pi_i}(x_*)\right\|^2\right]. \tag{32}$$

We now bound the second term in the right-hand side of (32). First, using Lemma 1, we get

$$
\begin{aligned}
\mathbb{E}\left[\left\|\sum_{i=k}^{n-1} \nabla f_{\pi_i}(x_*)\right\|^2\right] &= (n-k)^2 \mathbb{E}\left[\left\|\frac{1}{n-k}\sum_{i=k}^{n-1} \nabla f_{\pi_i}(x_*)\right\|^2\right] \\
&= (n-k)^2 \frac{k}{(n-k)(n-1)}\sigma_*^2 \\
&= \frac{k(n-k)}{n-1}\sigma_*^2.
\end{aligned}
$$

Next, by summing this for $k$ from 0 to $n-1$, we obtain

$$\sum_{k=0}^{n-1}\mathbb{E}\left[\left\|\sum_{i=k}^{n-1}\nabla f_{\pi_i}(x_*)\right\|^2\right]=\sum_{k=0}^{n-1}\frac{k(n-k)}{n-1}\sigma_*^2=\frac{1}{6}n(n+1)\sigma_*^2\leq\frac{n^2\sigma_*^2}{4},$$

where in the last step we also used $n\geq 2$. The result follows. ∎

### 8.4.2 Finding a per-epoch recursion

**Lemma 3.** Assume that functions $f_1,\ldots,f_n$ are convex and that Assumption 1 is satisfied. If Random Reshuffling (Algorithm 1) or Shuffle-Once (Algorithm 2) is run with a stepsize satisfying $\gamma\leq\frac{1}{\sqrt{2}Ln}$, then

$$\mathbb{E}\left[\|x_{t+1}-x_*\|^2\right]\leq\mathbb{E}\left[\|x_t-x_*\|^2\right]-2\gamma n\mathbb{E}\left[f(x_{t+1})-f_*\right]+\frac{\gamma^3Ln^2\sigma_*^2}{2}.$$

*Proof.* Define the sum of gradients used in the $t$-th epoch as $g_t\stackrel{\text{def}}{=}\sum_{i=0}^{n-1}\nabla f_{\pi_i}(x_t^i)$. We will use $g_t$ to relate the iterates $x_t$ and $x_{t+1}$. By definition of $x_{t+1}$, we can write

$$x_{t+1}=x_t^n=x_t^{n-1}-\gamma\nabla f_{\pi_{n-1}}(x_t^{n-1})=\cdots=x_t^0-\gamma\sum_{i=0}^{n-1}\nabla f_{\pi_i}(x_t^i).$$

Further, since $x_t^0=x_t$, we see that $x_{t+1}=x_t-\gamma g_t$, which leads to

$$\begin{aligned}
\|x_t-x_*\|^2=\|x_{t+1}+\gamma g_t-x_*\|^2&=\|x_{t+1}-x_*\|^2+2\gamma\langle g_t,x_{t+1}-x_*\rangle+\gamma^2\|g_t\|^2\\
&\geq\|x_{t+1}-x_*\|^2+2\gamma\langle g_t,x_{t+1}-x_*\rangle\\
&=\|x_{t+1}-x_*\|^2+2\gamma\sum_{i=0}^{n-1}\left\langle\nabla f_{\pi_i}(x_t^i),x_{t+1}-x_*\right\rangle.
\end{aligned}$$

Observe that for any $i$, we have the following decomposition

$$\begin{aligned}
\left\langle\nabla f_{\pi_i}(x_t^i),x_{t+1}-x_*\right\rangle&=[f_{\pi_i}(x_{t+1})-f_{\pi_i}(x_*)]+[f_{\pi_i}(x_*)-f_{\pi_i}(x_t^i)-\langle\nabla f_{\pi_i}(x_t^i),x_t^i-x_*\rangle]\\
&\quad-[f_{\pi_i}(x_{t+1})-f_{\pi_i}(x_t^i)-\langle\nabla f_{\pi_i}(x_t^i),x_{t+1}-x_t^i\rangle]\\
&=[f_{\pi_i}(x_{t+1})-f_{\pi_i}(x_*)]+D_{f_{\pi_i}}(x_*,x_t^i)-D_{f_{\pi_i}}(x_{t+1},x_t^i). \qquad (33)
\end{aligned}$$

Summing the first quantity in (33) over $i$ from 0 to $n-1$ gives

$$\sum_{i=0}^{n-1}[f_{\pi_i}(x_{t+1})-f_{\pi_i}(x_*)]=n(f(x_{t+1})-f_*).$$

Now, we can bound the third term in the decomposition (33) using $L$-smoothness as follows:

$$D_{f_{\pi_i}}(x_{t+1},x_t^i)\leq\frac{L}{2}\|x_{t+1}-x_t^i\|^2.$$

By summing the right-hand side over $i$ from 0 to $n-1$ we get the forward deviation over an epoch $\mathcal{V}_t$, which we bound by Lemma 2 to get

$$\sum_{i=0}^{n-1}\mathbb{E}\left[D_{f_{\pi_i}}(x_{t+1},x_t^i)\right]\overset{(30)}{\leq}\frac{L}{2}\mathbb{E}\left[\mathcal{V}_t\right]\overset{(31)}{\leq}2\gamma^2L^2n^2\sum_{i=0}^{n-1}\mathbb{E}\left[D_{f_{\pi_i}}(x_*,x_t^i)\right]+\frac{\gamma^2Ln^2\sigma_*^2}{4}.$$

Therefore, we can lower-bound the sum of the second and the third term in (33) as

$$\begin{aligned}
\sum_{i=0}^{n-1}\mathbb{E}\left[D_{f_{\pi_i}}(x_*,x_t^i)-D_{f_{\pi_i}}(x_{t+1},x_t^i)\right]&\geq\sum_{i=0}^{n-1}\mathbb{E}\left[D_{f_{\pi_i}}(x_*,x_t^i)\right]-2\gamma^2L^2n^2\sum_{i=0}^{n-1}\mathbb{E}\left[D_{f_{\pi_i}}(x_*,x_t^i)\right]\\
&\quad-\frac{\gamma^2Ln^2\sigma_*^2}{4}\\
&\geq(1-2\gamma^2L^2n^2)\sum_{i=0}^{n-1}\mathbb{E}\left[D_{f_{\pi_i}}(x_*,x_t^i)\right]-\frac{\gamma^2Ln^2\sigma_*^2}{4}\\
&\geq-\frac{\gamma^2Ln^2\sigma_*^2}{4},
\end{aligned}$$

where in the third inequality we used that $\gamma \leq \frac{1}{\sqrt{2}Ln}$ and that $D_{f_{\pi_i}}(x_*, x_t^i)$ is nonnegative. Plugging this back into the lower-bound on $\|x_t - x_*\|^2$ yields

$$\mathbb{E}\left[\|x_t - x_*\|^2\right] \geq \mathbb{E}\left[\|x_{t+1} - x_*\|^2\right] + 2\gamma n \mathbb{E}\left[f(x_{t+1}) - f_*\right] - \frac{\gamma^3 Ln^2\sigma_*^2}{2}.$$

Rearranging the terms gives the result. ∎

## 8.5 Proof of Theorem 2

*Proof.* We can use Lemma 3 and strong convexity to obtain

$$\mathbb{E}\left[\|x_{t+1} - x_*\|^2\right] \leq \mathbb{E}\left[\|x_t - x_*\|^2\right] - 2\gamma n \mathbb{E}\left[f(x_{t+1}) - f_*\right] + \frac{\gamma^3 Ln^2\sigma_*^2}{2}$$

$$\overset{(13)}{\leq} \mathbb{E}\left[\|x_t - x_*\|^2\right] - \gamma n\mu \mathbb{E}\left[\|x_{t+1} - x_*\|^2\right] + \frac{\gamma^3 Ln^2\sigma_*^2}{2},$$

whence

$$\mathbb{E}\left[\|x_{t+1} - x_*\|^2\right] \leq \frac{1}{1 + \gamma\mu n}\left(\mathbb{E}\left[\|x_t - x_*\|^2\right] + \frac{\gamma^3 Ln^2\sigma_*^2}{2}\right)$$

$$= \frac{1}{1 + \gamma\mu n}\mathbb{E}\left[\|x_t - x_*\|^2\right] + \frac{1}{1 + \gamma\mu n}\frac{\gamma^3 Ln^2\sigma_*^2}{2}$$

$$\leq \left(1 - \frac{\gamma\mu n}{2}\right)\mathbb{E}\left[\|x_t - x_*\|^2\right] + \frac{\gamma^3 Ln^2\sigma_*^2}{2}.$$

Recursing for $T$ iterations, we get that the final iterate satisfies

$$\mathbb{E}\left[\|x_T - x_*\|^2\right] \leq \left(1 - \frac{\gamma\mu n}{2}\right)^T \|x_0 - x_*\|^2 + \frac{\gamma^3 Ln^2\sigma_*^2}{2}\left(\sum_{j=0}^{T-1}\left(1 - \frac{\gamma\mu n}{2}\right)^j\right)$$

$$\leq \left(1 - \frac{\gamma\mu n}{2}\right)^T \|x_0 - x_*\|^2 + \frac{\gamma^3 Ln^2\sigma_*^2}{2}\left(\frac{2}{\gamma\mu n}\right)$$

$$= \left(1 - \frac{\gamma\mu n}{2}\right)^T \|x_0 - x_*\|^2 + \gamma^2\kappa n\sigma_*^2. \qquad \blacksquare$$

## 8.6 Proof of Theorem 3

*Proof.* We start with Lemma 3, which states that the following inequality holds:

$$\mathbb{E}\left[\|x_{t+1} - x_*\|^2\right] \leq \mathbb{E}\left[\|x_t - x_*\|^2\right] - 2\gamma n \mathbb{E}\left[f(x_{t+1}) - f(x_*)\right] + \frac{\gamma^3 Ln^2\sigma_*^2}{2}.$$

Rearranging the result leads to

$$2\gamma n \mathbb{E}\left[f(x_{t+1}) - f(x_*)\right] \leq \mathbb{E}\left[\|x_t - x_*\|^2\right] - \mathbb{E}\left[\|x_{t+1} - x_*\|^2\right] + \frac{\gamma^3 Ln^2\sigma_*^2}{2}.$$

Summing these inequalities for $t = 0, 1, \ldots, T-1$ gives

$$2\gamma n \sum_{t=0}^{T-1} \mathbb{E}\left[f(x_{t+1}) - f(x_*)\right] \leq \sum_{t=0}^{T-1}\left(\mathbb{E}\left[\|x_t - x_*\|^2\right] - \mathbb{E}\left[\|x_{t+1} - x_*\|^2\right]\right) + \frac{\gamma^3 Ln^2\sigma_*^2 T}{2}$$

$$= \|x_0 - x_*\|^2 - \mathbb{E}\left[\|x_T - x_*\|^2\right] + \frac{\gamma^3 Ln^2\sigma_*^2 T}{2}$$

$$\leq \|x_0 - x_*\|^2 + \frac{\gamma^3 Ln^2\sigma_*^2 T}{2},$$

and dividing both sides by $2\gamma nT$, we get

$$\frac{1}{T}\sum_{t=0}^{T-1} \mathbb{E}\left[f(x_{t+1}) - f(x_*)\right] \leq \frac{\|x_0 - x_*\|^2}{2\gamma nT} + \frac{\gamma^2 Ln\sigma_*^2}{4}.$$

Finally, using convexity of $f$, the average iterate $\hat{x}_T \overset{\text{def}}{=} \frac{1}{T}\sum_{t=1}^{T} x_t$ satisfies

$$\mathbb{E}\left[f(\hat{x}_T) - f(x_*)\right] \leq \frac{1}{T}\sum_{t=1}^{T} \mathbb{E}\left[f(x_t) - f(x_*)\right] \leq \frac{\|x_0 - x_*\|^2}{2\gamma nT} + \frac{\gamma^2 Ln\sigma_*^2}{4}. \qquad \blacksquare$$

## 8.7 Proof of complexity

**Corollary 2.** Under the same conditions as Theorem 3, choose the stepsize

$$\gamma = \min\left\{\frac{1}{\sqrt{2}Ln}, \left(\frac{\|x_0 - x_*\|^2}{Ln^2T\sigma_*^2}\right)^{1/3}\right\}.$$

Then

$$\mathbb{E}\left[f(\hat{x}_T) - f(x_*)\right] \leq \frac{L\|x_0 - x_*\|^2}{\sqrt{2}T} + \frac{3L^{1/3}\|x_0 - x_*\|^{4/3}\sigma_*^{2/3}}{4n^{1/3}T^{2/3}}.$$

We can guarantee $\mathbb{E}\left[f(\hat{x}_T) - f(x_*)\right] \leq \varepsilon^2$ provided that the total number of iterations satisfies

$$Tn \geq \frac{2\|x_0 - x_*\|^2\sqrt{Ln}}{\varepsilon^2}\max\left\{\sqrt{2Ln}, \frac{\sigma_*}{\varepsilon}\right\}.$$

*Proof.* We start with the guarantee of Theorem 3:

$$\mathbb{E}\left[f(\hat{x}_T) - f(x_*)\right] \leq \frac{\|x_0 - x_*\|^2}{2\gamma nT} + \frac{\gamma^2 Ln\sigma_*^2}{4}. \tag{34}$$

We now have two cases depending on the stepsize:

- **Case 1**: If $\gamma = \frac{1}{\sqrt{2}Ln} \leq \left(\frac{\|x_0-x_*\|^2}{Ln^2T\sigma_*^2}\right)^{1/3}$, then plugging this $\gamma$ into (34) gives

$$\mathbb{E}\left[f(\hat{x}_T) - f(x_*)\right] \leq \frac{L\|x_0 - x_*\|^2}{\sqrt{2}T} + \frac{\gamma^2 Ln\sigma_*^2}{4}$$

$$\leq \frac{L\|x_0 - x_*\|^2}{\sqrt{2}T} + \left(\frac{\|x_0 - x_*\|^2}{Ln^2T\sigma_*^2}\right)^{2/3}\frac{Ln\sigma_*^2}{4}$$

$$= \frac{L\|x_0 - x_*\|^2}{\sqrt{2}T} + \frac{L^{1/3}\sigma_*^{2/3}\|x_0 - x_*\|^{4/3}}{4n^{1/3}T^{2/3}}. \tag{35}$$

- **Case 2**: If $\gamma = \left(\frac{\|x_0-x_*\|^2}{Ln^2T\sigma_*^2}\right)^{1/3} \leq \frac{1}{\sqrt{2}Ln}$, then plugging this $\gamma$ into (34) gives

$$\mathbb{E}\left[f(\hat{x}_T) - f(x_*)\right] \leq \frac{L^{1/3}\|x_0 - x_*\|^{4/3}\sigma_*^{2/3}}{2n^{1/3}T^{2/3}} + \frac{L^{1/3}\sigma_*^{2/3}\|x_0 - x_*\|^{4/3}}{4n^{1/3}T^{2/3}}$$

$$= \frac{3L^{1/3}\|x_0 - x_*\|^{4/3}\sigma_*^{2/3}}{4n^{1/3}T^{2/3}}. \tag{36}$$

Combining (35) and (36), we see that in both cases we have

$$\mathbb{E}\left[f(\hat{x}_T) - f(x_*)\right] \leq \frac{L\|x_0 - x_*\|^2}{\sqrt{2}T} + \frac{3L^{1/3}\|x_0 - x_*\|^{4/3}\sigma_*^{2/3}}{4n^{1/3}T^{2/3}}.$$

Translating this to sample complexity, we can guarantee that $\mathbb{E}\left[f(\hat{x}_T) - f(x_*)\right] \leq \varepsilon^2$ provided

$$nT \geq \frac{2\|x_0 - x_*\|^2\sqrt{Ln}}{\varepsilon^2}\max\left\{\sqrt{Ln}, \frac{\sigma_*}{\varepsilon}\right\}. \qquad \blacksquare$$

# 9 Proofs for non-convex objectives (Section 3.3)

## 9.1 Proof of Proposition 2

*Proof.* This proposition is a special case of Lemma 3 in (Khaled and Richtárik, 2020) and we prove it here for completeness. Let $x \in \mathbb{R}^d$. We start with (16) (which does not require convexity) applied to each $f_i$:

$$\|\nabla f_i(x)\|^2 \le 2L \left( f_i(x) - f_i^* \right).$$

Averaging, we derive

$$\frac{1}{n} \sum_{i=1}^{n} \|\nabla f_i(x)\|^2 \le 2L \left( f(x) - \frac{1}{n} \sum_{i=1}^{n} f_i^* \right)$$

$$= 2L \left( f(x) - f_* \right) + 2L \left( f_* - \frac{1}{n} \sum_{i=1}^{n} f_i^* \right).$$

Note that because $f_*$ is the infimum of $f(\cdot)$ and $\frac{1}{n} \sum_{i=1}^{n} f_i^*$ is a lower bound on $f$ then $f_* - \frac{1}{n} \sum_{i=1}^{n} f_i^* \ge 0$. We may now use the variance decomposition

$$\frac{1}{n} \sum_{i=1}^{n} \|\nabla f_i(x) - \nabla f(x)\|^2 \stackrel{(21)}{=} \frac{1}{n} \sum_{i=1}^{n} \|\nabla f_i(x)\|^2 - \|\nabla f(x)\|^2$$

$$\le \frac{1}{n} \sum_{i=1}^{n} \|\nabla f_i(x)\|^2$$

$$\le 2L \left( f(x) - f_* \right) + 2L \left( f_* - \frac{1}{n} \sum_{i=1}^{n} f_i^* \right).$$

It follows that Assumption 2 holds with $A = L$ and $B^2 = 2L \left( f_* - \frac{1}{n} \sum_{i=1}^{n} f_i^* \right)$. ∎

## 9.2 Finding a per-epoch recursion

For this subsection and the rest of this section, we need to define the following quantity:

**Definition 4.** For Algorithm 1 we define the *backward per-epoch deviation* at timestep $t$ by

$$V_t \stackrel{\text{def}}{=} \frac{1}{n} \sum_{i=1}^{n} \left\| x_t^i - x_t \right\|^2.$$

We will study the convergence of Algorithm 1 for non-convex objectives as follows: we first derive a per-epoch recursion that involves $V_t$ in Lemma 4, then we show that $V_t$ can be bounded using smoothness and probability theory in Lemma 5, and finally combine these two to prove Theorem 4.

**Lemma 4.** Suppose that Assumption 1 holds. Then for iterates $x_t$ generated by Algorithm 1 with stepsize $\gamma \le \frac{1}{Ln}$, we have

$$f(x_{t+1}) \le f(x_t) - \frac{\gamma n}{2} \|\nabla f(x_t)\|^2 + \frac{\gamma L^2}{2} V_t, \tag{37}$$

where $V_t$ is defined as in Definition 4.

*Proof.* Our approach for establishing this lemma is similar to that of (Nguyen et al., 2020, Theorem 1), which we became aware of in the course of preparing this manuscript. Recall that $x_{t+1} = x_t - \gamma g_t$, where $g_t = \sum_{i=0}^{n-1} \nabla f_{\pi_i}(x_t^i)$. Using $L$-smoothness of $f$, we get

$$f(x_{t+1}) \stackrel{(15)}{\le} f(x_t) + \langle \nabla f(x_t), x_{t+1} - x_t \rangle + \frac{L}{2} \|x_{t+1} - x_t\|^2$$

$$= f(x_t) - \gamma n \left\langle \nabla f(x_t), \frac{g_t}{n} \right\rangle + \frac{\gamma^2 L n^2}{2} \left\| \frac{g_t}{n} \right\|^2$$

$$\stackrel{(11)}{=} f(x_t) - \frac{\gamma n}{2} \left( \|\nabla f(x_t)\|^2 + \left\| \frac{g_t}{n} \right\|^2 - \left\| \nabla f(x_t) - \frac{g_t}{n} \right\|^2 \right) + \frac{\gamma^2 L n^2}{2} \left\| \frac{g_t}{n} \right\|^2$$

$$= f(x_t) - \frac{\gamma n}{2} \|\nabla f(x_t)\|^2 - \frac{\gamma n}{2} \left( 1 - L \gamma n \right) \left\| \frac{g_t}{n} \right\|^2 + \frac{\gamma n}{2} \left\| \nabla f(x_t) - \frac{g_t}{n} \right\|^2. \tag{38}$$

By assumption, we have $\gamma \leq \frac{1}{Ln}$, and hence $1 - L\gamma n \geq 0$. Using this in (38), we get

$$f(x_{t+1}) \leq f(x_t) - \frac{\gamma n}{2}\|\nabla f(x_t)\|^2 + \frac{\gamma n}{2}\left\|\nabla f(x_t) - \frac{g_t}{n}\right\|^2. \tag{39}$$

For the last term in (39), we note

$$
\begin{aligned}
\left\|\nabla f(x_t) - \frac{g_t}{n}\right\|^2 &= \left\|\frac{1}{n}\sum_{i=0}^{n-1}\left[\nabla f_{\pi_i}(x_t) - \nabla f_{\pi_i}(x_t^i)\right]\right\|^2 \\
&\overset{(18)}{\leq} \frac{1}{n}\sum_{i=0}^{n-1}\left\|\nabla f_{\pi_i}(x_t) - \nabla f_{\pi_i}(x_t^i)\right\|^2 \\
&\overset{(14)}{\leq} \frac{1}{n}\sum_{i=0}^{n-1}L^2\|x_t - x_t^i\|^2 = \frac{L^2}{n}V_t.
\end{aligned} \tag{40}
$$

Plugging in (40) into (39) yields the lemma's claim. ∎

### 9.3 Bounding the backward per-epoch deviation

**Lemma 5.** Suppose that Assumption 1 holds (with each $f_i$ possibly non-convex) and that Algorithm 1 is used with a stepsize $\gamma \leq \frac{1}{2Ln}$. Then

$$\mathbb{E}_t\left[V_t\right] \leq \gamma^2 n^3\|\nabla f(x_t)\|^2 + \gamma^2 n^2 \sigma_t^2, \tag{41}$$

where $V_t$ is defined as in Definition 4 and $\sigma_t^2 \overset{\text{def}}{=} \frac{1}{n}\sum_{j=1}^{n}\|\nabla f_j(x_t) - \nabla f(x_t)\|^2$.

*Proof.* Let us fix any $k \in [1, n-1]$ and find an upper bound for $\mathbb{E}_t\left[\|x_t^k - x_t\|^2\right]$. First, note that

$$x_t^k = x_t - \gamma\sum_{i=0}^{k-1}\nabla f_{\pi_i}(x_t^i).$$

Therefore, by Young's inequality, Jensen's inequality and gradient Lipschitzness

$$
\begin{aligned}
\mathbb{E}_t\left[\|x_t^k - x_t\|^2\right] &= \gamma^2\mathbb{E}_t\left[\left\|\sum_{i=0}^{k-1}\nabla f_{\pi_i}(x_t^i)\right\|^2\right] \\
&\overset{(12)}{\leq} 2\gamma^2\mathbb{E}_t\left[\left\|\sum_{i=0}^{k-1}\left(\nabla f_{\pi_i}(x_t^i) - \nabla f_{\pi_i}(x_t)\right)\right\|^2\right] + 2\gamma^2\mathbb{E}_t\left[\left\|\sum_{i=0}^{k-1}\nabla f_{\pi_i}(x_t)\right\|^2\right] \\
&\overset{(19)}{\leq} 2\gamma^2 k\sum_{i=0}^{k-1}\mathbb{E}_t\left[\left\|\nabla f_{\pi_i}(x_t^i) - \nabla f_{\pi_i}(x_t)\right\|^2\right] + 2\gamma^2\mathbb{E}_t\left[\left\|\sum_{i=0}^{k-1}\nabla f_{\pi_i}(x_t)\right\|^2\right] \\
&\overset{(14)}{\leq} 2\gamma^2 L^2 k\sum_{i=0}^{k-1}\mathbb{E}_t\left[\|x_t^i - x_t\|^2\right] + 2\gamma^2\mathbb{E}_t\left[\left\|\sum_{i=0}^{k-1}\nabla f_{\pi_i}(x_t)\right\|^2\right].
\end{aligned}
$$

Let us bound the second term. For any $i$ we have $\mathbb{E}_t\left[\nabla f_{\pi_i}(x_t)\right] = \nabla f(x_t)$, so using Lemma 1 (with vectors $\nabla f_{\pi_0}(x_t), \nabla f_{\pi_1}(x_t), \ldots, \nabla f_{\pi_{k-1}}(x_t)$) we obtain

$$
\begin{aligned}
\mathbb{E}_t\left[\left\|\sum_{i=0}^{k-1}\nabla f_{\pi_i}(x_t)\right\|^2\right] &\overset{(20)}{=} k^2\|\nabla f(x_t)\|^2 + k^2\mathbb{E}_t\left[\left\|\frac{1}{k}\sum_{i=0}^{k-1}(\nabla f_{\pi_i}(x_t) - \nabla f(x_t))\right\|^2\right] \\
&\overset{(22)}{\leq} k^2\|\nabla f(x_t)\|^2 + \frac{k(n-k)}{n-1}\sigma_t^2.
\end{aligned}
$$

where $\sigma_t^2 \overset{\text{def}}{=} \frac{1}{n}\sum_{j=1}^{n}\|\nabla f_j(x_t) - \nabla f(x_t)\|^2$. Combining the produced bounds yields

$$\mathbb{E}_t\left[\left\|x_t^k - x_t\right\|^2\right] \leq 2\gamma^2 L^2 k \sum_{i=0}^{k-1}\mathbb{E}_t\left[\left\|x_t^i - x_t\right\|^2\right] + 2\gamma^2 k^2 \|\nabla f(x_t)\|^2 + 2\gamma^2 \frac{k(n-k)}{n-1}\sigma_t^2$$

$$\leq 2\gamma^2 L^2 k \mathbb{E}\left[V_t\right] + 2\gamma^2 k^2 \|\nabla f(x_t)\|^2 + 2\gamma^2 \frac{k(n-k)}{n-1}\sigma_t^2,$$

whence

$$\mathbb{E}\left[V_t\right] = \sum_{k=0}^{n-1}\mathbb{E}_t\left[\left\|x_t^k - x_t\right\|^2\right]$$

$$\leq \gamma^2 L^2 n(n-1)\mathbb{E}\left[V_t\right] + \frac{1}{3}\gamma^2(n-1)n(2n-1)\|\nabla f(x_t)\|^2 + \frac{1}{3}\gamma^2 n(n+1)\sigma_t^2.$$

Since $\mathbb{E}\left[V_t\right]$ appears in both sides of the equation, we rearrange and use that $\gamma \leq \frac{1}{2Ln}$ by assumption, which leads to

$$\mathbb{E}\left[V_t\right] \leq \frac{4}{3}(1 - \gamma^2 L^2 n(n-1))\mathbb{E}\left[V_t\right]$$

$$\leq \frac{4}{9}\gamma^2(n-1)n(2n-1)\|\nabla f(x_t)\|^2 + \frac{4}{9}\gamma^2 n(n+1)\sigma_t^2$$

$$\leq \gamma^2 n^3 \|\nabla f(x_t)\|^2 + \gamma^2 n^2 \sigma_t^2. \qquad \blacksquare$$

### 9.4 A lemma for solving the non-convex recursion

**Lemma 6.** Suppose that there exist constants $a, b, c \geq 0$ and nonnegative sequences $(s_t)_{t=0}^{T}, (q_t)_{t=0}^{T}$ such that for any $t$ satisfying $0 \leq t \leq T$ we have the recursion

$$s_{t+1} \leq (1 + a)\, s_t - bq_t + c. \tag{42}$$

Then, the following holds:

$$\min_{t=0,\dots,T-1} q_t \leq \frac{(1 + a)^T}{bT}s_0 + \frac{c}{b}. \tag{43}$$

*Proof.* The first part of the proof (for $a > 0$) is a distillation of the recursion solution in Lemma 2 of Khaled and Richtárik (2020) and we closely follow their proof. Define

$$w_t \overset{\text{def}}{=} \frac{1}{(1 + a)^{t+1}}.$$

Note that $w_t(1 + a) = w_{t-1}$ for all $t$. Multiplying both sides of (42) by $w_t$,

$$w_t s_{t+1} \leq (1 + a)\, w_t s_t - bw_t q_t + cw_t = w_{t-1}s_t - bw_t q_t + cw_t.$$

Rearranging, we get $bw_t q_t \leq w_{t-1}s_t - w_t s_{t+1} + cw_t$. Summing up as $t$ varies from 0 to $T - 1$ and noting that the sum telescopes leads to

$$\sum_{t=0}^{T-1}bw_t q_t \leq \sum_{t=0}^{T-1}(w_{t-1}s_t - w_t s_{t+1}) + c\sum_{t=0}^{T-1}w_t$$

$$= w_0 s_0 - w_{T-1}s_T + c\sum_{t=0}^{T-1}w_t$$

$$\leq w_0 s_0 + c\sum_{t=0}^{T-1}w_t.$$

Let $W_T = \sum_{t=0}^{T-1}w_t$. Dividing both sides by $W_T$, we get

$$\frac{1}{W_T}\sum_{t=0}^{T-1}bw_t q_t \leq \frac{w_0 s_0}{W_T} + c. \tag{44}$$

Note that the left-hand side of (44) satisfies

$$b \min_{t=0,\dots,T-1} q_t \leq \frac{1}{W_T} \sum_{t=0}^{T-1} b w_t q_t. \tag{45}$$

For the right-hand side of (44), we have

$$W_T = \sum_{t=0}^{T-1} w_t \geq T \min_{t=0,\dots,T-1} w_t = T w_{T-1} = \frac{T}{(1+a)^T}. \tag{46}$$

Substituting with (46) in (45) and dividing both sides by $b$, we finally get

$$\min_{t=0,\dots,T-1} q_t \leq \frac{(1+a)^T}{bT} s_0 + \frac{c}{b}. \qquad \blacksquare$$

## 9.5 Proof of Theorem 4

*Proof.* Taking expectation in Lemma 4 and then using Lemma 5, we have that for any $t \in \{0, 1, \dots, T-1\}$,

$$
\begin{aligned}
\mathbb{E}_t \left[ f(x_{t+1}) \right] &\overset{(37)}{\leq} f(x_t) - \frac{\gamma n}{2} \|\nabla f(x_t)\|^2 + \frac{\gamma L^2}{2} \mathbb{E}_t \left[ V_t \right] \\
&\overset{(41)}{\leq} f(x_t) - \frac{\gamma n}{2} \|\nabla f(x_t)\|^2 + \frac{\gamma L^2}{2} \left( \gamma^2 n^3 \|\nabla f(x_t)\|^2 + \gamma^2 n^2 \sigma_t^2 \right) \\
&= f(x_t) - \frac{\gamma n}{2} \left( 1 - \gamma^2 L^2 n^2 \right) \|\nabla f(x_t)\|^2 + \frac{\gamma^3 L^2 n^2 \sigma_t^2}{2}.
\end{aligned}
$$

Let $\delta_t \overset{\text{def}}{=} f(x_t) - f_*$. Adding $-f_*$ to both sides and using Assumption 2,

$$
\begin{aligned}
\mathbb{E}_t \left[ \delta_{t+1} \right] &\leq \delta_t - \frac{\gamma n}{2} \left( 1 - \gamma^2 L^2 n^2 \right) \|\nabla f(x_t)\|^2 + \frac{\gamma^3 L^2 n^2 \sigma_t^2}{2} \\
&\leq \left( 1 + \gamma^3 A L^2 n^2 \right) \delta_t - \frac{\gamma n}{2} \left( 1 - \gamma^2 L^2 n^2 \right) \|\nabla f(x_t)\|^2 + \frac{\gamma^3 L^2 n^2 B^2}{2}.
\end{aligned}
$$

Taking unconditional expectations in the last inequality and using that by assumption on $\gamma$ we have $1 - \gamma^2 L^2 n^2 \geq \frac{1}{2}$, we get the estimate

$$\mathbb{E}\left[\delta_{t+1}\right] \leq \left(1 + \gamma^3 A L^2 n^2\right) \mathbb{E}\left[\delta_t\right] - \frac{\gamma n}{4} \mathbb{E}\left[\|\nabla f(x_t)\|^2\right] + \frac{\gamma^3 L^2 n^2 B^2}{2}. \tag{47}$$

Comparing (42) with (47) verifies that the conditions of Lemma 6 are readily satisfied. Applying the lemma, we get

$$\min_{t=0,\dots,T-1} \mathbb{E}\left[\|\nabla f(x_t)\|^2\right] \leq \frac{4\left(1 + \gamma^3 A L^2 n^2\right)^T}{\gamma n T} \left(f(x_0) - f_*\right) + 2\gamma^2 L^2 n B^2.$$

Using that $1 + x \leq \exp(x)$ and that the stepsize $\gamma$ satisfies $\gamma \leq \left(A L^2 n^2 T\right)^{-1/3}$, we have

$$\left(1 + \gamma^3 A L^2 n^2\right)^T \leq \exp\left(\gamma^3 A L^2 n^2 T\right) \leq \exp(1) \leq 3.$$

Using this in the previous bound, we finally obtain

$$\min_{t=0,\dots,T-1} \mathbb{E}\left[\|\nabla f(x_t)\|^2\right] \leq \frac{12\left(f(x_0) - f_*\right)}{\gamma n T} + 2\gamma^2 L^2 n B^2. \qquad \blacksquare$$

## 9.6 Proof of complexity

**Corollary 3.** Choose the stepsize $\gamma$ as

$$\gamma = \min\left\{ \frac{1}{2Ln}, \frac{1}{A^{1/3} L^{2/3} n^{2/3} T^{1/3}}, \frac{\varepsilon}{2L\sqrt{nB}} \right\}.$$

Then the minimum gradient norm satisfies

$$\min_{t=0,\ldots,T-1} \mathbb{E}\left[\|\nabla f(x_t)\|^2\right] \leq \varepsilon^2$$

provided the total number of iterations satisfies

$$Tn \geq \frac{48\delta_0 L\sqrt{n}}{\varepsilon^2} \max\left\{\sqrt{n}, \frac{\sqrt{6\delta_0 A}}{\varepsilon}, \frac{B}{\varepsilon}\right\}.$$

*Proof.* From Theorem 4

$$\min_{t=0,\ldots,T-1} \mathbb{E}\left[\|\nabla f(x_t)\|^2\right] \leq \frac{12\left(f(x_0) - f_*\right)}{\gamma nT} + 2\gamma^2 L^2 nB^2.$$

Note that by condition on the stepsize $2L^2\gamma^2 nB^2 \leq \varepsilon^2/2$, hence

$$\min_{t=0,\ldots,T-1} \mathbb{E}\left[\|\nabla f(x_t)\|^2\right] \leq \frac{12\left(f(x_0) - f_*\right)}{\gamma nT} + \frac{\varepsilon^2}{2}.$$

Thus, to make the squared gradient norm smaller than $\varepsilon^2$ we require

$$\frac{12\left(f(x_0) - f_*\right)}{\gamma nT} \leq \frac{\varepsilon^2}{2},$$

or equivalently

$$nT \geq \frac{24\left(f(x_0) - f_*\right)}{\varepsilon^2\gamma} = \frac{24\delta_0}{\varepsilon^2} \max\left\{2Ln, \left(AL^2 n^2 T\right)^{1/3}, \frac{2L\sqrt{n}B}{\varepsilon}\right\}, \qquad (48)$$

where $\delta_0 \overset{\text{def}}{=} f(x_0) - f_*$ and where we plugged in the value of the stepsize $\gamma$ we use. Note that $nT$ appears on both sides in the second term in the maximum in (48), hence we can cancel out and simplify:

$$nT \geq \frac{24\delta_0}{\varepsilon^2}(AL^2 n^2 T)^{1/3} \iff nT \geq \frac{(24\delta_0)^{3/2}L\sqrt{An}}{\varepsilon^3}.$$

Using this simplified bound in (48) we obtain that $\min_{t=0,\ldots,T-1} \mathbb{E}\left[\|\nabla f(x_t)\|^2\right] \leq \varepsilon^2$ provided

$$nT \geq \frac{48\delta_0 L\sqrt{n}}{\varepsilon^2} \max\left\{\sqrt{n}, \frac{\sqrt{6\delta_0 A}}{\varepsilon}, \frac{B}{\varepsilon}\right\}. \qquad \blacksquare$$

## 10    Convergence results for IG

In this section we present results that are extremely similar to the previously obtained bounds for RR and SO. For completeness, we also provide a full description of IG in Algorithm 3.

---

**Algorithm 3** Incremental Gradient (IG)

---

**Input:** Stepsize $\gamma > 0$, initial vector $x_0 = x_0^0 \in \mathbb{R}^d$, number of epochs $T$
1: **for** epochs $t = 0, 1, \ldots, T-1$ **do**
2:     **for** $i = 0, 1, \ldots, n-1$ **do**
3:         $x_t^{i+1} = x_t^i - \gamma \nabla f_{i+1}(x_t^i)$
4:     $x_{t+1} = x_t^n$

---

**Theorem 5.** Suppose that Assumption 1 is satisfied. Then we have the following results for the Incremental Gradient method:

- **If each $f_i$ is $\mu$-strongly convex**: if $\gamma \leq \frac{1}{L}$, then

$$\|x_T - x_*\|^2 \leq (1 - \gamma\mu)^{nT} \|x_0 - x_*\|^2 + \frac{\gamma^2 Ln^2\sigma_*^2}{\mu}.$$

By carefully choosing the stepsize as in Corollary 1, we see that this result implies that IG has sample complexity $\tilde{\mathcal{O}}\left(\kappa + \frac{\sqrt{\kappa}n\sigma_*}{\mu\sqrt{\varepsilon}}\right)$ in order to reach a point $\tilde{x}$ with $\|\tilde{x} - x_*\|^2 \leq \varepsilon$.

- **If $f$ is $\mu$-strongly convex and each $f_i$ is convex**: if $\gamma \le \frac{1}{\sqrt{2}nL}$, then

$$\|x_T - x_*\|^2 \le \left(1 - \frac{\gamma\mu n}{2}\right)^T \|x_0 - x_*\|^2 + 2\gamma^2\kappa n^2\sigma_*^2.$$

  Using the same approach for choosing the stepsize as Corollary 1, we see that IG in this setting reaches an $\varepsilon$-accurate solution after $\tilde{\mathcal{O}}\left(n\kappa + \frac{\sqrt{\kappa}n\sigma_*}{\mu\sqrt{\varepsilon}}\right)$ individual gradient accesses.

- **If each $f_i$ is convex**: if $\gamma \le \frac{1}{\sqrt{2}nL}$, then

$$f(\hat{x}_T) - f(x_*) \le \frac{\|x_0 - x_*\|^2}{2\gamma nT} + \frac{\gamma^2 Ln^2\sigma_*^2}{2},$$

  where $\hat{x}_T \overset{\text{def}}{=} \frac{1}{T}\sum_{t=1}^{T} x_t$. Choosing the stepsize $\gamma = \min\left\{\frac{1}{\sqrt{2}nL}, \frac{\sqrt{\varepsilon}}{\sqrt{L}n\sigma_*}\right\}$, then the average of iterate generated by IG is an $\varepsilon$-accurate solution (i.e., $f(\hat{x}_T) - f(x_*) \le \varepsilon$) provided that the total number of iterations satisfies

$$nT \ge \frac{\|x_0 - x_*\|^2}{\varepsilon}\max\left\{\sqrt{8}nL, \frac{\sqrt{L}\sigma_*n}{\sqrt{\varepsilon}}\right\}.$$

- **If each $f_i$ is possibly non-convex**: if Assumption 2 holds with constants $A, B \ge 0$ and $\gamma \le \min\left\{\frac{1}{\sqrt{8}nL}, \frac{1}{(4L^2n^3AT)^{1/3}}\right\}$, then

$$\min_{t=0,\ldots,T-1} \|\nabla f(x_t)\|^2 \le \frac{12\left(f(x_0) - f_*\right)}{\gamma nT} + 8\gamma^2L^2n^2B^2.$$

  Using an approach similar to Corollary 3, we can establish that IG reaches a point with gradient norm less than $\varepsilon$ provided that the total number of iterations exceeds

$$nT \ge \frac{48\left(f(x_0) - f_*\right)Ln}{\varepsilon^2}\max\left\{\sqrt{2}, \frac{\sqrt{24\left(f(x_0) - f_*\right)A}}{\varepsilon}, \frac{2B}{\varepsilon}\right\}.$$

The proof of Thoerem 5 is given in the rest of the section, but first we briefly discuss the convergence rates and the relation of the result on strongly convex objectives to the lower bound of Safran and Shamir (2020).

**Discussion of the convergence rates.** A brief comparison between the sample complexities given for IG in Theorem 5 and those given for RR (in Table 1) reveals that IG has similar rates to RR but with a worse dependence on $n$ in the variance term (the term associated with $\sigma_*$ in the convex case and $B$ in the non-convex case), in particular IG is worse by a factor of $\sqrt{n}$. This difference is significant in the large-scale machine learning regime, where the number of data points $n$ can be on the order of thousands to millions.

**Discussion of existing lower bounds.** Safran and Shamir (2020) give the lower bound (in a problem with $\kappa = 1$)

$$\|x_T - x_*\|^2 = \Omega\left(\frac{\sigma_*^2}{\mu^2T^2}\right).$$

This implies a sample complexity of $\mathcal{O}\left(\frac{n\sigma_*}{\mu\sqrt{\varepsilon}}\right)$, which matches our upper bound (up to an extra iteration and log factors) in the case each $f_i$ is strongly convex and $\kappa = 1$.

## 10.1 Preliminary Lemmas for Theorem 5

### 10.1.1 Two lemmas for convex objectives

**Lemma 7.** Consider the iterates of Incremental Gradient (Algorithm 3). Suppose that functions $f_1, \ldots, f_n$ are convex and that Assumption 1 is satisfied. Then it holds

$$\sum_{k=0}^{n-1} \left\|x_t^k - x_{t+1}\right\|^2 \le 4\gamma^2 Ln^2\sum_{i=0}^{n-1} D_{f_{i+1}}(x_*, x_t^i) + 2\gamma^2n^3\sigma_*^2, \tag{49}$$

where $\sigma_*^2$ is the variance at the optimum given by $\sigma_*^2 \stackrel{\text{def}}{=} \frac{1}{n} \sum_{i=1}^{n} \|\nabla f_i(x_*)\|^2$.

*Proof.* The proof of this Lemma is similar to that of Lemma 2 but with a worse dependence on the variance term, since there is no randomness in IG. Fix any $k \in \{0, \ldots, n-1\}$. It holds by definition

$$x_t^k - x_{t+1} = \gamma \sum_{i=k}^{n-1} \nabla f_{i+1}(x_t^i) = \gamma \sum_{i=k}^{n-1} (\nabla f_{i+1}(x_t^i) - \nabla f_{i+1}(x_*)) + \gamma \sum_{i=k}^{n-1} \nabla f_{i+1}(x_*).$$

Applying Young's inequality to the sums above yields

$$
\begin{aligned}
\|x_t^k - x_{t+1}\|^2 &\stackrel{(12)}{\leq} 2\gamma^2 \left\| \sum_{i=k}^{n-1} (\nabla f_{i+1}(x_t^i) - \nabla f_{i+1}(x_*)) \right\|^2 + 2\gamma^2 \left\| \sum_{i=k}^{n-1} \nabla f_{i+1}(x_*) \right\|^2 \\
&\stackrel{(19)}{\leq} 2\gamma^2 n \sum_{i=k}^{n-1} \|\nabla f_{i+1}(x_t^i) - \nabla f_{i+1}(x_*)\|^2 + 2\gamma^2 \left\| \sum_{i=k}^{n-1} \nabla f_{i+1}(x_*) \right\|^2 \\
&\stackrel{(17)}{\leq} 4\gamma^2 Ln \sum_{i=k}^{n-1} D_{f_{i+1}}(x_*, x_t^i) + 2\gamma^2 \left\| \sum_{i=k}^{n-1} \nabla f_{i+1}(x_*) \right\|^2 \\
&\leq 4\gamma^2 Ln \sum_{i=0}^{n-1} D_{f_{i+1}}(x_*, x_t^i) + 2\gamma^2 \left\| \sum_{i=k}^{n-1} \nabla f_{i+1}(x_*) \right\|^2 .
\end{aligned}
$$

Summing up,

$$\sum_{k=0}^{n-1} \left\| x_t^k - x_{t+1} \right\|^2 \leq 4\gamma^2 Ln^2 \sum_{i=0}^{n-1} D_{f_{i+1}}(x_*, x_t^i) + 2\gamma^2 \sum_{k=0}^{n-1} \left\| \sum_{i=k}^{n-1} \nabla f_{i+1}(x_*) \right\|^2 . \tag{50}$$

We now bound the second term in (50). We have

$$
\begin{aligned}
\sum_{k=0}^{n-1} \left\| \sum_{i=k}^{n-1} \nabla f_{i+1}(x_*) \right\|^2 &\stackrel{(19)}{\leq} \sum_{k=0}^{n-1} (n-k) \sum_{i=k}^{n-1} \|\nabla f_{i+1}(x_*)\|^2 \\
&\leq \sum_{k=0}^{n-1} (n-k) \sum_{i=0}^{n-1} \|\nabla f_{i+1}(x_*)\|^2 \\
&= \sum_{k=0}^{n-1} (n-k) \, n\sigma_*^2 = \frac{n^2(n+1)}{2} \sigma_*^2 \leq n^3 \sigma_*^2 . \tag{51}
\end{aligned}
$$

Using (51) in (50), we derive

$$\sum_{k=0}^{n-1} \left\| x_t^k - x_{t+1} \right\|^2 \leq 4\gamma^2 Ln^2 \sum_{i=0}^{n-1} D_{f_{i+1}}(x_*, x_t^i) + 2\gamma^2 n^3 \sigma_*^2 . \qquad \blacksquare$$

**Lemma 8.** Assume the functions $f_1, \ldots, f_n$ are convex and that Assumption 1 is satisfied. If Algorithm 3 is run with a stepsize $\gamma \leq \frac{1}{\sqrt{2}Ln}$, then

$$\|x_{t+1} - x_*\|^2 \leq \|x_t - x_*\|^2 - 2\gamma n \left( f(x_{t+1}) - f(x_*) \right) + \gamma^3 Ln^3 \sigma_*^2 .$$

*Proof.* The proof for this lemma is identical to Lemma 3 but with the estimate of Lemma 7 used for $\sum_{i=0}^{n-1} \left\| x_t^i - x_{t+1} \right\|^2$ instead of Lemma 2. We only include it for completeness. Define the sum of gradients used in the $t$-th epoch as $g_t \stackrel{\text{def}}{=} \sum_{i=0}^{n-1} \nabla f_{i+1}(x_t^i)$. By definition of $x_{t+1}$, we have $x_{t+1} = x_t - \gamma g_t$. Using this,

$$
\begin{aligned}
\|x_t - x_*\|^2 = \|x_{t+1} + \gamma g_t - x_*\|^2 &= \|x_{t+1} - x_*\|^2 + 2\gamma \langle g_t, x_{t+1} - x_* \rangle + \gamma^2 \|g_t\|^2 \\
&\geq \|x_{t+1} - x_*\|^2 + 2\gamma \langle g_t, x_{t+1} - x_* \rangle \\
&= \|x_{t+1} - x_*\|^2 + 2\gamma \sum_{i=0}^{n-1} \left\langle \nabla f_{i+1}(x_t^i), x_{t+1} - x_* \right\rangle .
\end{aligned}
$$

For any $i$ we have the following decomposition

$$\langle \nabla f_{i+1}(x_t^i), x_{t+1} - x_* \rangle = [f_{i+1}(x_{t+1}) - f_{i+1}(x_*)] \tag{52}$$
$$+ [f_{i+1}(x_*) - f_{i+1}(x_t^i) - \langle \nabla f_{i+1}(x_t^i), x_t^i - x_* \rangle]$$
$$- [f_{i+1}(x_{t+1}) - f_{i+1}(x_t^i) - \langle \nabla f_{i+1}(x_t^i), x_{t+1} - x_t^i \rangle]$$
$$= [f_{i+1}(x_{t+1}) - f_{i+1}(x_*)] + D_{f_{i+1}}(x_*, x_t^i) - D_{f_{i+1}}(x_{t+1}, x_t^i). \tag{53}$$

Summing the first quantity in (53) over $i$ from $0$ to $n-1$ gives

$$\sum_{i=0}^{n-1} [f_{i+1}(x_{t+1}) - f_{i+1}(x_*)] = n(f(x_{t+1}) - f_*).$$

Now let us work out the third term in the decomposition (53) using $L$-smoothness,

$$D_{f_{i+1}}(x_{t+1}, x_t^i) \leq \frac{L}{2} \|x_{t+1} - x_t^i\|^2.$$

We next sum the right-hand side over $i$ from $0$ to $n-1$ and use Lemma 7

$$\sum_{i=0}^{n-1} D_{f_{i+1}}(x_{t+1}, x_t^i) \quad \leq \quad \frac{L}{2} \sum_{i=0}^{n-1} \|x_{t+1} - x_t^i\|^2$$
$$\overset{(49)}{\leq} \quad 2\gamma^2 L^2 n^2 \sum_{i=0}^{n-1} D_{f_{i+1}}(x_*, x_t^i) + \gamma^2 L n^3 \sigma_*^2.$$

Therefore, we can lower-bound the sum of the second and the third term in (53) as

$$\sum_{i=0}^{n-1} (D_{f_{i+1}}(x_*, x_t^i) - D_{f_{i+1}}(x_{t+1}, x_t^i)) \geq \sum_{i=0}^{n-1} D_{f_{i+1}}(x_*, x_t^i)$$
$$- \left( 2\gamma^2 L^2 n^2 \sum_{i=0}^{n-1} D_{f_{i+1}}(x_*, x_t^i) - \gamma^2 L n^3 \sigma_*^2 \right)$$
$$= (1 - 2\gamma^2 L^2 n^2) \sum_{i=0}^{n-1} D_{f_{i+1}}(x_*, x_t^i) - \gamma^2 L n^3 \sigma_*^2$$
$$\geq -\gamma^2 L n^3 \sigma_*^2,$$

where in the third inequality we used that $\gamma \leq \frac{1}{\sqrt{2}Ln}$ and that $D_{f_{i+1}}(x_*, x_t^i)$ is nonnegative. Plugging this back into the lower-bound on $\|x_t - x_*\|^2$ yields

$$\|x_t - x_*\|^2 \geq \|x_{t+1} - x_*\|^2 + 2\gamma n \left( f(x_{t+1}) - f_* \right) - \gamma^3 L n^3 \sigma_*^2.$$

Rearranging the terms gives the result. ∎

### 10.1.2   A lemma for non-convex objectives

**Lemma 9.** Suppose that Assumption 1 holds. Suppose that Algorithm 3 is used with a stepsize $\gamma > 0$ such that $\gamma \leq \frac{1}{2Ln}$. Then we have,

$$\sum_{i=1}^{n} \|x_t^i - x_t\|^2 \leq 4\gamma^2 n^3 \|\nabla f(x_t)\|^2 + 4\gamma^2 n^3 \sigma_t^2, \tag{54}$$

where $\sigma_t^2 \overset{\text{def}}{=} \frac{1}{n} \sum_{j=1}^{n} \|\nabla f_j(x_t) - \nabla f(x_t)\|^2$.

*Proof.* Let $i \in \{1, 2, \ldots, n\}$. Then we can bound the deviation of a single iterate as,

$$\|x_t^i - x_t\|^2 = \left\| x_t^0 - \gamma \sum_{j=0}^{i-1} \nabla f_{j+1}(x_t^j) - x_t \right\|^2 \quad = \quad \gamma^2 \left\| \sum_{j=0}^{i-1} \nabla f_{j+1}(x_t^j) \right\|^2$$
$$\overset{(19)}{\leq} \quad \gamma^2 i \sum_{j=0}^{i-1} \left\| \nabla f_{j+1}(x_t^j) \right\|^2.$$

Because $i \leq n$, we have

$$\left\|x_t^i - x_t\right\|^2 \leq \gamma^2 i \sum_{j=0}^{i-1} \left\|\nabla f_{i+1}(x_t^j)\right\|^2 \leq \gamma^2 n \sum_{j=0}^{i-1} \left\|\nabla f_{i+1}(x_t^j)\right\|^2 \leq \gamma^2 n \sum_{j=0}^{n-1} \left\|\nabla f_{i+1}(x_t^j)\right\|^2. \quad (55)$$

Summing up allows us to estimate $V_t$:

$$
\begin{aligned}
V_t &= \sum_{i=1}^{n} \left\|x_t^i - x_t\right\|^2 \\
&\overset{(55)}{\leq} \sum_{i=1}^{n} \left( \gamma^2 n \sum_{j=0}^{n-1} \left\|\nabla f_{j+1}(x_t^j)\right\|^2 \right) \\
&= \gamma^2 n^2 \sum_{j=0}^{n-1} \left\|\nabla f_{j+1}(x_t^j)\right\|^2 \\
&\overset{(12)}{\leq} 2\gamma^2 n^2 \sum_{j=0}^{n-1} \left( \left\|\nabla f_{j+1}(x_t^j) - \nabla f_{i+1}(x_t)\right\|^2 + \left\|\nabla f_{j+1}(x_t)\right\|^2 \right) \\
&= 2\gamma^2 n^2 \sum_{j=0}^{n-1} \left\|\nabla f_{j+1}(x_t^j) - \nabla f_{j+1}(x_t)\right\|^2 + 2\gamma^2 n^2 \sum_{j=0}^{n-1} \left\|\nabla f_{j+1}(x_t)\right\|^2. \quad (56)
\end{aligned}
$$

For the first term in (56) we can use the smoothness of individual losses and that $x_t^0 = x_t$:

$$\sum_{j=0}^{n-1} \left\|\nabla f_{j+1}(x_t^j) - \nabla f_{j+1}(x_t)\right\|^2 \overset{(14)}{\leq} L^2 \sum_{j=0}^{n-1} \left\|x_t^j - x_t\right\|^2 = L^2 \sum_{j=1}^{n-1} \left\|x_t^j - x_t\right\|^2 = L^2 V_t. \quad (57)$$

The second term in (56) is a sum over all the individual gradient evaluated at the same point $x_t$. Hence, we can drop the permutation subscript and then use the variance decomposition:

$$
\begin{aligned}
\sum_{j=0}^{n-1} \|\nabla f_{i+1}(x_t)\|^2 &= \sum_{j=1}^{n} \|\nabla f_j(x_t)\|^2 \\
&\overset{(21)}{=} n\|\nabla f(x_t)\|^2 + \sum_{j=1}^{n} \|\nabla f_j(x_t) - \nabla f(x_t)\|^2 \\
&= n\|\nabla f(x_t)\|^2 + n\sigma_t^2. \quad (58)
\end{aligned}
$$

We can then use (57) and (58) in (56),

$$V_t \leq 2\gamma^2 L^2 n^2 V_t + 2\gamma^2 n^3 \|\nabla f(x_t)\|^2 + 2\gamma^2 n^3 \sigma_t^2.$$

Since $V_t$ shows up in both sides of the equation, we can rearrange to obtain

$$\left(1 - 2\gamma^2 L^2 n^2\right) V_t \leq 2\gamma^2 n^3 \|\nabla f(x_t)\|^2 + 2\gamma^2 n^3 \sigma_t^2.$$

If $\gamma \leq \frac{1}{2Ln}$, then $1 - 2\gamma^2 L^2 n^2 \geq \frac{1}{2}$ and hence

$$V_t \leq 4\gamma^2 n^3 \|\nabla f(x_t)\|^2 + 4\gamma^2 n^3 \sigma_t^2. \qquad \blacksquare$$

## 10.2 Proof of Theorem 5

*Proof.* • **If each $f_i$ is $\mu$-strongly convex**: The proof follows that of Theorem 1. Define

$$x_*^i = x_* - \gamma \sum_{j=0}^{i-1} \nabla f_{j+1}(x_*).$$

First, we have

$$\|x_t^{i+1} - x_*^{i+1}\|^2$$
$$= \|x_t^i - x_*^i\|^2 - 2\gamma \langle \nabla f_{i+1}(x_t^i) - \nabla f_{i+1}(x_*), x_t^i - x_*^i \rangle + \gamma^2 \|\nabla f_{i+1}(x_t^i) - \nabla f_{i+1}(x_*)\|^2.$$

Using the same three-point decomposition as Theorem 1 and strong convexity, we have

$$-\left\langle \nabla f_{i+1}(x_t^i) - \nabla f_{i+1}(x_*), x_t^i - x_*^i \right\rangle = -D_{f_{i+1}}(x_*^i, x_t^i) - D_{f_{i+1}}(x_t^i, x_*) + D_{f_{i+1}}(x_*^i, x_*)$$
$$\leq -\frac{\mu}{2}\left\|x_t^i - x_*^i\right\|^2 - D_{f_{i+1}}(x_t^i, x_*) + D_{f_{i+1}}(x_*^i, x_*).$$

Using smoothness and convexity

$$\frac{1}{2L}\left\|\nabla f_{i+1}(x_t^i) - \nabla f_{i+1}(x_*)\right\|^2 \leq D_{f_{i+1}}(x_t^i, x_*).$$

Plugging in the last two inequalities into the recursion, we get

$$\left\|x_t^{i+1} - x_*^{i+1}\right\|^2 \leq (1 - \gamma\mu)\left\|x_t^i - x_*^i\right\|^2 - 2\gamma\left(1 - \gamma L\right)D_{f_{i+1}}(x_t^i, x_*) + 2\gamma D_{f_{i+1}}(x_*^i, x_*).$$
$$\leq (1 - \gamma\mu)\left\|x_t^i - x_*^i\right\|^2 + 2\gamma D_{f_{i+1}}(x_*^i, x_*). \tag{59}$$

For the last Bregman divergence, we have

$$D_{f_{i+1}}(x_*^i, x_*) \overset{(15)}{\leq} \frac{L}{2}\left\|x_*^i - x_*\right\|^2$$
$$= \frac{\gamma^2 L}{2}\left\|\sum_{j=0}^{i-1} \nabla f_{j+1}(x_*)\right\|^2$$
$$\overset{(19)}{\leq} \frac{\gamma^2 L i}{2}\sum_{j=0}^{i-1}\left\|\nabla f_{j+1}(x_*)\right\|^2$$
$$= \frac{\gamma^2 L i n}{2}\sigma_*^2 \leq \frac{\gamma^2 L n^2}{2}\sigma_*^2.$$

Plugging this into (59), we get

$$\left\|x_t^{i+1} - x_*^{i+1}\right\|^2 \leq (1 - \gamma\mu)\left\|x_t^i - x_*^i\right\|^2 + \gamma^3 L n^2 \sigma_*^2.$$

We recurse and then use that $x_*^n = x_*$, $x_{t+1} = x_t^n$, and that $x_*^0 = x_*$, obtaining

$$\|x_{t+1} - x_*\|^2 = \|x_t^n - x_*^n\|^2 \leq (1 - \gamma\mu)^n\left\|x_t^0 - x_*^0\right\|^2 + \gamma^3 L n^2 \sigma_*^2 \sum_{j=0}^{n-1}(1 - \gamma\mu)^j$$
$$= (1 - \gamma\mu)^n\|x_t - x_*\|^2 + \gamma^3 L n^2 \sigma_*^2 \sum_{j=0}^{n-1}(1 - \gamma\mu)^j.$$

Recursing again,

$$\|x_T - x_*\|^2 \leq (1 - \gamma\mu)^{nT}\|x_0 - x_*\|^2 + \gamma^3 L n^2 \sigma_*^2 \sum_{j=0}^{n-1}(1 - \gamma\mu)^j \sum_{t=0}^{T-1}(1 - \gamma\mu)^{nt}$$
$$= (1 - \gamma\mu)^{nT}\|x_0 - x_*\|^2 + \gamma^3 L n^2 \sigma_*^2 \sum_{k=0}^{nT-1}(1 - \gamma\mu)^k$$
$$\leq (1 - \gamma\mu)^{nT}\|x_0 - x_*\|^2 + \frac{\gamma^3 L n^2 \sigma_*^2}{\gamma\mu}$$
$$= (1 - \gamma\mu)^{nT}\|x_0 - x_*\|^2 + \gamma^2 \kappa n^2 \sigma_*^2.$$

- **If $f$ is $\mu$-strongly convex and each $f_i$ is convex**: the proof is identical to that of Theorem 2 but using Lemma 8 instead of Lemma 3, and we omit it for brevity.

- **If each $f_i$ is convex**: the proof is identical to that of Theorem 3 but using Lemma 8 instead of Lemma 3, and we omit it for brevity.

- **If each $f_i$ is possibly non-convex**: note that Lemma 4 also applies to IG without change, hence if $\gamma \leq \frac{1}{Ln}$ we have

$$f(x_{t+1}) \leq f(x_t) - \frac{\gamma n}{2}\|\nabla f(x_t)\|^2 + \frac{\gamma L^2}{2}\sum_{i=1}^{n}\left\|x_t - x_t^i\right\|^2.$$

We may then apply Lemma 9 to get for $\gamma \leq \frac{1}{2Ln}$

$$f(x_{t+1}) \leq f(x_t) - \frac{\gamma n}{2}\|\nabla f(x_t)\|^2 + \frac{\gamma L^2}{2}\left(4\gamma^2 n^3\|\nabla f(x_t)\|^2 + 4\gamma^2 n^3 \sigma_t^2\right)$$
$$= f(x_t) - \frac{\gamma n}{2}\left(1 - 4\gamma^2 L^2 n^2\right)\|\nabla f(x_t)\|^2 + 2\gamma^3 L^2 n^3 \sigma_t^2.$$

Using that $\gamma \leq \frac{1}{\sqrt{8}Ln}$ and subtracting $f_*$ from both sides, we derive

$$f(x_{t+1}) - f_* \leq (f(x_t) - f_*) - \frac{\gamma n}{4}\|\nabla f(x_t)\|^2 + 2\gamma^3 L^2 n^3 \sigma_t^2.$$

Using Assumption 2, we get

$$f(x_{t+1}) - f_* \leq \left(1 + 4\gamma^3 L^2 A n^3\right)(f(x_t) - f_*) - \frac{\gamma n}{4}\|\nabla f(x_t)\|^2 + 2\gamma^3 L^2 n^3 B^2. \qquad (60)$$

Applying Lemma 6 to (60), thus, gives

$$\min_{t=0,\ldots,T-1}\|\nabla f(x_t)\|^2 \leq \frac{4\left(1 + 4\gamma^3 L^2 A n^3\right)^T}{\gamma n T}(f(x_0) - f_*) + 8\gamma^2 L^2 n^2 B^2. \qquad (61)$$

Note that by our assumption on the stepsize, $4\gamma^3 L^2 A n^3 T \leq 1$, hence,

$$\left(1 + 4\gamma^3 L^2 A n^3\right)^T \leq \exp\left(4\gamma^3 L^2 A n^3 T\right) \leq \exp(1) \leq 3.$$

It remains to use this in (61). ∎