[Reviews · NeurIPS 2020]

Review 1

Summary and Contributions: The paper considers Random Reshuffling paradigm for stochastic gradient descent used for empirical risk minimization. Here, in each epoch the data points are processed in the order given by a uniformly random permutation. There have been many analyses prior to this work which have established convergence rates for RR and have shown that it performs better than with-replacement SGD used in practice. In the case of strongly convex and smooth objectives, this work removes several assumptions made in prior works - like bounded gradients and bounded variance and improves the convergence rates by a factor of condition number. They also extend the analysis of RR to non-strongly convex but smooth functions which are the first results, to the best of my knowledge, to show that RR converges faster than with-replacement sampling albeit with a worse dependence on n, the number of samples available. The analysis is general enough to be extended to the case of Shuffle Once paradigm, where the data is shuffled only once, to obtain sharp bounds on the convergence rates.

Strengths: 1. The technical content, as understood from the main body of the paper, appears to be sounds and interesting. I really like the idea of considering Bregman divergence from the optimum point, since this appears to be a natural notion of distance. 2. The analysis of RR in the case of non-strongly convex functions is novel and in my opinion, one of the strongest contributions of this work. 3. The sharp analysis of Shuffle Once paradigm is novel and is a significant contribution since this is used heavily in practice. ======== The authors have addressed most of my concerns with the rebuttal. I maintain my score.

Weaknesses: 1. Unlike the works HaoChen and Sra and Nagaraj et.al, this work uses the fact that all component functions f_i are mu strongly convex. 2. The authors need to explain why removing some of the assumptions like bounded variance and bounded gradients is an important contribution via. solid examples. 3. The quantity sigma^{*} being finite also implies that all the gradients are finite via. smoothness property of the functions f_i and gives a natural upper bound.

Correctness: I read through the main body of the work and skimmed through the supplementary material. The claims appear to be correct based on my reading.It would be helpful to the reader if the authors included some technical content - which is sufficient to broadly explain the proof techniques used - in the main body of the work.

Clarity: Yes. The paper is well written.

Relation to Prior Work: The work cites most of the relevant prior work in this field. They need to do a better job of distinguishing the assumptions in existing works from the assumptions in the paper (see point 1. in weaknesses section).

Reproducibility: Yes

Additional Feedback:


Review 2

Summary and Contributions: Post-rebuttal update: To authors: Thank you for your responses. I maintain that the paper should be accepted. ========= This paper contributes a number of results around the performance of random reshuffling (RR), shuffle once (SO) and incremental gradient (IG) for the optimization of finite sums that are strongly convex, convex or non-convex. The question around why RR and SO seems to perform much better than SGD in modern applications had so far not been answered rigorously. The authors resolve many of those important questions. - A critical component in the proposed style of analysis is a different way to define variance, called shuffling variance. - The authors use that new definition to give a convergence result of the case when component functions are individually strongly convex. They point out how their bound matches a lower bound by Safran and Shamir for a specific case where kappa=1. - The authors point out that for RR and SO, using mini batches of size tau results to a reduction of variance of 1/tau^2. Figure 2 beautifully depicts this point. - The result extends to the case of convex component functions, however in that case a much smaller step size is necessary O(1/Ln) vs O(1/L). - Finally the authors provide results for smooth, non-convex finite sums. Simple experiments validate many of the produced pieces of intuition regarding the relative performance of RR/SO/SGD/IG.

Strengths: - The paper reads effortlessly. Results and presentation are easy and accessible, despite the fact that they involve non-trivial technical work and contributions. - The topic is relevant to ML and the previously open questions important. - The answers are clear and corroborated by both theory and experiments.

Weaknesses: Almost none. I would appreciate it if the authors would spent a little more time explaining the intuition behind their definition of 'shuffling variance'. I believe that I get the idea (and the math works out) but it's such an important piece of this type of analysis. Explaining it a bit more would have great educational value.

Correctness: Yes

Clarity: The paper is written super clearly. At the top of my stack when it comes to clear presentation and quality of writing and organization.

Relation to Prior Work: The the best of my knowledge there are no pieces of relevant recent or older work missing. The improvement over the previous state of the art results is clear.

Reproducibility: Yes

Additional Feedback: Great paper!


Review 3

Summary and Contributions: 1. The paper gives convergence rates for the widely used algorithms Random Reshuffling, Shuffle Once and Iterative Gradient. Although these are used a lot in practice, these have remained theoretically difficult to analyze. 2. The paper improves upon the prior work on Random Reshuffling by removing large epoch requirements. 3. This is the first paper to give tight upper bound on convergence rates for Shuffle Once algorithm, under some strong assumptions.

Strengths: 1. Theorem 1 gives an upper bound for Shuffle Once algorithm, matching the lower bound given by Safran and Shamir (2020). However, the paper assumes strong convexity of each component function, which I believe makes the analysis much easier (see the Weakness section below for details). 2. Theorem 2 removes the large epoch requirements for Random Reshuffle from the results of Nagaraj et al. (2019). 3. The analysis in the paper seems to be much simpler than the Wasserstein distance and coupling based analysis of prior work (Nagaraj et al. (2019)). In particular, the paper introduces 'shuffling variance' and build on the intuition that within an epoch, these algorithms try to optimize a different function than the objective function. Dheeraj Nagaraj, Prateek Jain, and Praneeth Netrapalli. "SGD without replacement: Sharper rates for general smooth convex functions." International Conference on Machine Learning. 2019. Itay Safran, and Ohad Shamir. "How good is SGD with random shuffling?." arXiv preprint arXiv:1908.00045 (2019).

Weaknesses: 1. The abstract claims to remove the small step size requirements of prior work. However, to attain a good convergence rate (Corollary 1) the main theorems (Theorems 1 and 2) need a small step size, similar to previous works. In fact Safran and Shamir (2020) show that convergence is only possible for step size $O(1/n)$. Please modify the claims accordingly. 2. The dependence of error on condition number is indeed reduced from $\kappa^2$ (Nagaraj et al. (2019)) to $\kappa$. However, the dependence on $\mu$ has worsened. In particular, Nagaraj et al. (2019) give an error rate of $\kappa^2/\mu$ (ignoring dependence on $n$ and $T$), whereas this paper gives a dependence of $\kappa/\mu^2$ and depending on whether $L$ is larger than or smaller than 1, one result is better than the other. Thus, the claim of reducing dependence on condition number should be modified to also state this. 3. The result of Theorem 3 says that RR beats SGD only after $\Omega(n)$ epochs, which seems really large. This can probably be improved. 4. Theorem 1 assumes that the individual functions are strongly convex. This is a really strong assumption which, in my opinion, the previous papers have deliberately avoided. This assumption is violated even if one of the component functions is not strongly convex. For example, even if one of the component functions is linear, then this assumption would not hold. In deep neural networks for instance, it can happen that in the neighborhood of a minima, the objective function behaves like a strongly convex function, but one (or more) of the component functions behaves linearly in the same neighborhood. I also believe that this assumption makes the analysis much more easier because now the sum of every subset of the individual functions is also strongly convex. Itay Safran, and Ohad Shamir. "How good is SGD with random shuffling?." arXiv preprint arXiv:1908.00045. 2019. Dheeraj Nagaraj, Prateek Jain, and Praneeth Netrapalli. "SGD without replacement: Sharper rates for general smooth convex functions." International Conference on Machine Learning. 2019. ----Post author feedback comments---- 1. The author feedback says that Theorem 2 can be extended to Shuffle Once, which seems to be true looking at the proof. Thus, now Theorem 2 is much stronger than Theorem 1 in terms of assumptions and in particular, point 4 above (regarding individual strong convexity) does not apply to Theorem 2. I believe this is a really good result and I'm increasing the score due to this reason. Note that I still believe that point 4 is a significant concern for Theorem 1. 2. The author feedback for point 2 seems satisfactory. I overlooked the fact that Nagaraj et al. (2019) talk about function sub-optimality and this paper presents results on distance to optimizer. I apologize for the oversight. A detailed comparison of Theorem 2 and (Nagaraj et al. (2019)) should be done so that the reader can easily see the improvement. 3. For point 1, I disagree with the author feedback. I think that from the viewpoints of both theory and modern practical applications, achieving 0 error (asymptotically) is of central importance. The authors have agreed to add a discussion about this. 4. I still think the literature review and comparison with lower bounds and other upper bounds isn't done properly (For example, there is no lower bound comparison for Theorem 2; and neither Theorem 1 nor Theorem 2 have been compared with the upper bound of Nagaraj et al. (2019)). The authors have agreed to improve upon this. Also I've just noticed that the dependence of sample complexities on $\epsilon$ is different in the two theorems (lines 180 and 201), shouldn't theorem 2 also have $\sqrt{\epsilon}$?

Correctness: The proofs and the empirical methodology seems correct.

Clarity: The paper is well written and intuitions are well explained.

Relation to Prior Work: The literature review is somewhat poor. The main results pertain to Random Reshuffling and yet the recent related works have been squeezed into 2 lines at the end of the literature review section. There is no discussion and the only comparison is in the form of a table. The results of Nagaraj et al. (2019) should be discussed in detail because they were the first to achieve optimal rates for Random Reshuffle, and one of the main contributions of this paper is removing the large epoch requirements from Nagaraj et al. (2019). Also, all the relevant upper bounds should be compared in detail with the lower bounds given by Safran and Shamir (2019) and Rajput et al. (2020). Dheeraj Nagaraj, Prateek Jain, and Praneeth Netrapalli. "SGD without replacement: Sharper rates for general smooth convex functions." International Conference on Machine Learning. 2019. Itay Safran, and Ohad Shamir. "How good is SGD with random shuffling?." arXiv preprint arXiv:1908.00045 (2019). Shashank Rajput, Anant Gupta, and Dimitris Papailiopoulos. "Closing the convergence gap of SGD without replacement." arXiv preprint arXiv:2002.10400 (2020).

Reproducibility: Yes

Additional Feedback: Adding proof sketches of the theorems to the main text of the paper would help the reader easily understand the proof techniques and intuitions. line 6: "strongly convex and smooth functions" should be changed to "smooth strongly convex" functions, to ensure that the reader does not think that these assumptions are also removed for non convex smooth functions.


Review 4

Summary and Contributions: The paper studies the random reshuffling (RR) algorithm where the data is shuffled once at the beginning of epoch and the shuffle once (SO) algorithm where the data is shuffled only once at the beginning. They give convergence results for these algorithms for finite sum minimization problem. They improve the dependence of kappa and number of functions in the convergence rate for these two algorithms when the functions are strongly convex. They also give convergence results for the case when the functions are convex and non-convex and improve better dependence of certain parameters.

Strengths: The paper studies the convergence rates of RR and SO algorithms which is an interesting topic and relevant to the NeurIPS community. As the paper mentions, these algorithms are commonly used in practice and understanding them is an important problem. For the case where each function is strongly convex, they improve the dependence on n and kappa as compared to previous works for the RR algorithm which matches the known lower bounds. They also remove the assumption that each of the functions is Lipschitz which was required in previous works. They also get improved results for non strongly convex and non convex functions under weaker assumptions. Their convergence results for the strongly convex case also extends to the SO algorithm which are the first convergence results for this algorithm as mentioned in the paper. A few questions for the authors: 1) They mention that the their convergence results for RR for strongly convex case matches the known lower bounds on line 178. Here, they discuss the case of kappa=1? Does the polynomial dependence on kappa also match the lower bounds? 2) Can the authors discuss if their results for the non strongly case can be improved or are there known lower bounds? 3) Their results for SO and RR match for the strongly convex case. However, the analysis for RR for non strongly convex and non convex case does not apply to SO. Can the authors please comment on if a similar analysis might be possible for SO?

Weaknesses: The paper claims that they derive a new notion of variance which helps explain why RR performs better than SGD from the beginning. I find this discussion a bit confusing (see point 1 below). Also, is it possible to compute this analytically for some simple problem settings and show that it is indeed smaller as compared to the original notion of variance. A few questions for the authors: 1) The authors mention on line 189 that which algorithm is faster boils down to which variance is smaller and they discuss how it can be smaller when the batch size is large. As mentioned in proposition 1, \sigma^2_shuffle is lower bounded by a term. If I assume the condition number to be 1 and put step size gamma=1/L, we get \sigma^2_shuffle \geq n\sigma^*/8 where n is the number of functions. Hence, it seems like \sigma^2_shuffle is much bigger than \sigma^* unless the step size is very small even if the batch size is some constant. Can the authors clarify this? 2) I think it would be useful to write the SGD convergence rate as in equation 4 to explicitly see when RR is better than SGD. In particular, looking at lines 173 and 188, it is hard to know which is better unless we know the step sizes schedules are used in each of the algorithms. 3) For figure 2, how are the values \sigma^2_shuffle and \sigma^* estimated? And, what are the values of L and \mu that are used? -------------------------------------------------------------------------------- After reading the author response and other reviewers' comments, I have increased the rating since my concerns have been sufficiently addressed. The techniques in this paper are also interesting/novel as noted by other reviewers which I had overlooked before.

Correctness: The proofs seem correct at a first glance.

Clarity: The paper is well written. See comments above.

Relation to Prior Work: Yes, the paper discusses the related work clearly. One useful point to include would be to briefly discuss why earlier works required the Lipschitz assumption on individual functions and how they get rid of it for strongly convex and convex case.

Reproducibility: Yes

Additional Feedback:

[Author Response · NeurIPS 2020]

We thank the reviewers for their work and for the positive evaluation of our paper. Every reviewer noted that it is
well-written and reproducible. There seems to be no disagreement that the paper is of practical value (noted by R1, R2,
R4), relevant to the conference (R2, R4), and is generally on an interesting topic (R1, R2).

**To all reviewers.** We realized that we downplayed our contribution: Theorems 2 and 3 apply to Shuffle-Once without a
single change in the proofs, just as in Theorem 1. Thus, we also provided guarantees for SO without strong convexity.

**Reviewer 1.** 1.*"this work uses the fact that all component functions $f_i$ are $\mu$ strongly convex"* This assumption is not
strong because strong convexity typically comes from $\ell_2$ regularization or weight decay, which is normally used in each
function. Adding a small amount of regularization is also a common practice for numerical stability.
2. *"why removing some of the assumptions like bounded variance and bounded gradients is an important contribution"*
This assumption does not hold for any strongly convex function, which includes any convex objective with $\ell_2$ regular-
ization. It also doesn't hold for least squares, matrix factorization and neural networks (with smooth activations). These
problems are at the core of modern machine learning, and there are a lot of other similar objectives.
3. *"The quantity $\sigma_*$ being finite also implies that all the gradients are finite"* This is not true, $\sigma_*$ is always finite just
because $\nabla f_i(x_*)$ is a finite vector for any $i$, but globally the gradient is often not bounded (see the previous item).

**Reviewer 2.** We appreciate your support of our paper. Since accepted papers will be allowed to have an extra page, if
ours gets accepted, we will be happy to add discussion on the shuffling variance, which we agree will be educational.

**Reviewer 3.** 1. *"the main theorems (Theorems 1 and 2) need a small step size, similar to previous works. In fact
Safran and Shamir (2020) show that convergence is only possible for step size $O(1/n)$"* Firstly, we disagree about
Theorem 1—even with step size $\frac{1}{L}$ it guarantees convergence to a neighborhood. In practice, achieving machine
precision is sometimes not important, and a neighborhood would suffice. Furthermore, a substantial part of our work is
devoted to explaining why the neighborhood's size is small. Secondly, the argument of Safran and Shamir (2020) does
not show convergence is not possible with large stepsizes, indeed the proofs of Propositions 1 (p. 10, 2nd bullet point)
and Thm. 2 (p. 16, 4th bullet point) in their work both show that convergence proportional to $n$ and $T$ may be achieved
with a large stepsize. We will add these clarifications.
2. *"the dependence on $\mu$ has worsened. In particular, Nagaraj et al. (2019) give an error rate of $\kappa^2/\mu$"* This
comparison is not fair because Nagaraj et al. (2019) bound functional gap and our bound is for the distances. If we
apply $\|x - x_*\|^2 \le \frac{1}{\mu}(f(x) - f(x_*))$, we see that the bound of Nagaraj et al. gives rate $O(\kappa^2/\mu^2)$.
3. *"The result of Theorem 3 says that RR beats SGD only after $\Omega(n)$ epochs, which seems really large. This can
probably be improved."* This claim is somewhat speculative: we are unaware of any lower bounds in this setting. We
devoted a lot of time to tightening this bound and the complexity that we proved is better than any other available in the
literature for the same setting, so it is unclear why a better bound should be possible.
4. *"Theorem 1 assumes that the individual functions are strongly convex. This is a really strong assumption which, in
my opinion, the previous papers have deliberately avoided."* We disagree that it is such a strong assumption (see our
response to Reviewer 1). *"I also believe that this assumption makes the analysis much more easier because now the
sum of every subset of the individual functions is also strongly convex."* Our proof does not use that any sum of a subset
of the individual functions is strongly convex, so we do not see how this remark is relevant.
5. *"The literature review is somewhat poor."* We wrote a longer review but had to make it shorter because of the page
limit. If the paper gets accepted, we would be happy to use part of the allowed additional page to put more discussion.
6. *"Also, all the relevant upper bounds should be compared in detail with the lower bounds given by Safran and Shamir
(2019) and Rajput et al. (2020)."* We already compared the bounds with both papers, check page 6, in lines 177-185.

**Reviewer 4.** 1. *"Does the polynomial dependence on $\kappa$ also match the lower bounds?"* To our knowledge, there is no
lower bound with an explicit dependence on $\kappa$. As a sanity check, if $n = 1$, then $\sigma_* = 0$ and RR reduces to gradient
descent and we obtain the standard dependence on $\kappa$, so it is tight at least in this sense.
2. *"Can the authors discuss if their results for the non strongly case can be improved or are there known lower bounds?"*
We are not aware of any such lower bound. For SGD, recent work (Sebbouh et al. "On the convergence of the Stochastic
Heavy Ball Method") shows that momentum makes SGD faster when there is no strong convexity, despite SGD being
optimal in the strongly convex case. We think that momentum may improve the rate of RR as well.
3. *"Can the authors please comment on if a similar analysis might be possible for SO?"* We double-checked these
results and they turned out to apply to SO without any change in the proof.
4. *"it seems like $\sigma^2_{\text{shuffle}}$ is much bigger than $\sigma^2_*$ unless the step size is very small"* If $\kappa = 1$ and $\gamma = \frac{1}{L}$, then indeed the
shuffling variance can be in theory $n$ times bigger than $\sigma^2_*$ but we didn't observe this in our experiments.
5. *"I think it would be useful to write the SGD convergence rate as in equation 4"* We can add it: the first term will be
the same as in (4) and the second term would be $\frac{\sigma^2_*}{\mu^2 n T}$, where $nT$ is the total number of steps.
6. *"For figure 2, how are the values $\sigma^2_{\text{shuffle}}$ and $\sigma_*$ estimated? And, what are the values of $L$ and $\mu$ that are used?"* We
first obtained $x_*$ by running Nesterov's acceleration until machine precision, and then we estimated expectations by
randomly sampling permutations, and $\sigma_*$ was computed exactly.

[Meta-Review · NeurIPS 2020]

All reviewers agree on the novelty and importance of the results presented in the paper. This is a clear accept, and a solid contribution to the related literature on SGD without replacement.